# Applying the Jellium model to octacarbonyl metal complexes

Kun Wang[1,2], Chang Xu[1], Dan Li[1] & Longjiu Cheng [1,2✉]

The recently reported octacarbonyl metal complexes $M(CO)_8$ (M = Ca, Sr, Ba) feature interesting bonding structures. In these compounds, the bond order is 7, while accommodating 8 lone pairs of ligands in forming octa-coordinated complexes or ions. Here, by comparing $[Ba(CO)_8]^{2-}$ and metal clusters of $[BaBe_8]^{2-}$ analogically, we demonstrate that the Jellium model can not only be applied on metal clusters, but is also a useful tool to understand the electronic structures of $[M(CO)_8]^q$ (M, q = Ca, 2−; Sc, 1−; Ti, 0; V, 1+; Cr, 2+; Ba, 2−). By applying the Jellium model, we find that a 20-e model with the configuration $|1S^2|1P^6|1D^{10}|1F^2|$ is an appropriate description of the valence bonding structures of $M(CO)_8$ species, where each coordinative bond contains 7/8ths of the bonding orbitals and 1/8th non-bonding orbitals.

[1] Department of Chemistry, Anhui University, 230601 Hefei, Anhui, P.R. China. [2] AnHui Province Key Laboratory of Chemistry for Inorganic/Organic Hybrid Functionalized Materials, 230601 Hefei, Anhui, P.R. China. ✉email: clj@ustc.edu

The 18-electron rule is a very classic tool for us to understand the structures of a large amount of transition metal complexes[1]. Especially for the homoleptic carbonyl metal complexes, it is very successful to apply Dewar−Chatt−Duncanson (DCD) model[2] and 18-electron rule to explain the interactions between carbon monoxide and transition metals, such as the seven-coordinated carbonyl cations $[TM(CO)_7]^+$ (TM = $V^+$, $Nb^+$ and $Ta^+$) and the eight-coordinated carbonyl cations $[TM(CO)_8]^+$ (TM = $Sc^+$, $Y^+$ and $La^+$)[3–5]. However, in the recent synthesized alkaline earth metal complexes $M(CO)_8$ (M = Ca, Sr and Ba) or anions $[TM(CO)_8]^-$ (TM = $Sc^-$, $Y^-$ and $La^-$), only the valence electrons occupying metal−ligand bonding orbitals satisfy the DCD model and the 18-electron rule[6,7], including eight degenerate $\sigma_{CO \to TM}$ coordinative bonds and two $(n−1)d \to \pi^*$ backdonation bonds from alkali earth metal to the ligands[8].

There is an interesting inconsistency in understanding the electronic structures of octacarbonyl complexes $M(CO)_8$ (M = Ca, Sr and Ba) or anions $[TM(CO)_8]^-$ (TM = $Sc^-$, $Y^-$ and $La^-$) on the basis of different chemical bonding theories. It should be noted that all the structures adopt $O_h$ symmetry. Therefore, on the basis of valence bond (VB) theory[9], there are eight degenerate coordination bonds and two $\pi$-backdonation bonds between metal and ligands in $M(CO)_8$ complexes or ions, where eight carbonyl groups provide eight lone pairs (LPs) of electrons to the center metal. However, based on the hybrid orbital (HO) theory[10], only seven empty degenerate orbitals can be provided by the $d^3sp^3$ hybridized metal. Therefore, the bond orders are inconsistent on the basis of the two classic theories. It is difficult to arrange the 16 LPs in forming an eight-coordinated complex with the bond order of 7. As for $TM(CO)_8$ complex or ion, it has been hypothesized that the eight σ-type orbitals are contributed by both the ligands and the transition metal, where the transition metal possibly adopts a higher-level hybridization (such as the f-type polarization[3]) for bonding with the eight carbonyl groups. It is a reasonable viewpoint to understand the inconsistency of VB and HO theories, where the bond orders are both equal to 8.

Additionally, Zhou's research pointed out that $M(CO)_8$ complexes or ions have $O_h$ symmetry with the valence electron configuration of $a_{1g}^2t_{1u}^6t_{2g}^6a_{2u}^2e_g^2$, where the $a_{2u}$ orbital is explained as the ligand-only orbital, which satisfies the 18-electron rule perfectly[3,8]. The electrons in $a_{2u}$ orbital are stabilized by the field effect of the metal on the ligand cage[3,11]. Besides the ligand-only $a_{2u}$ orbital, the other nine MOs ($a_{1g} + 3t_{1u} + 3t_{2g} + 2e_g$) including seven σ-donation bonds ($a_{1g} + 3t_{1u} + 3t_{2g}$) and two π-backdonation bonds ($2e_g$) accommodate 18 valence electrons of $M(CO)_8$ complex perfectly[8], which is also quite reasonable to understand the bonding structures of $M(CO)_8$ complexes or ions.

On the basis of the experimental studies, all the ten valence orbitals ($a_{1g} + 3t_{1u} + 3t_{2g} + a_{2u} + 2e_g$) can be fulfilled with maximum 20 electrons for the $O_h$-symmetric octacarbonyl metal complexes or ions as singlet state. Therefore, it is attractive for us to understand the state of the ligand-only orbital ($a_{2u}$ orbital) essentially.

For another aspect, the octacarbonyl metal complex or ion is with $O_h$ symmetry, where the octa-coordinative field contributed by the positive center metal and eight ligands can be approximately viewed as a homogeneous spherical field or a Jellium model[12,13]. Therefore, we design a series of $O_h$-symmetric and singlet-state $[M(CO)_8]^q$ (M, q = Ca, 2−; Sc, 1−; Ti, 0; V, 1+; Cr, 2+; Ba, 2−) complex/ions theoretically to learn their bonding structures. Such $M^q(CO)_8$ complex/ions are 20-e closed-shell molecules based on the configuration[3], which exactly satisfy the magic stability of Jellium model[12,13]. We are curious that whether the ligand-only $a_{2u}$ orbital as a component participate in the eight coordinative bonds to form a 20-electron configuration for its valence electrons.

Here we first compare the molecular orbitals (MOs) of octacarbonyl metal cations $[Ba(CO)_8]^{2-}$ and metal cluster $[BaBe_8]^{2-}$ to demonstrate Jellium model is possibly appropriate for understanding the valence bonding orbitals. Then $[M(CO)_8]^q$ (M, q = Ca, 2−; Sc, 1−; Ti, 0; V, 1+; Cr, 2+) as the singlet "20e-superatom" models are designed and discussed, which are described by Jellium model successfully similar with that of $[Ba(CO)_8]^{2-}$ and $[BaBe_8]^{2-}$ with the configuration of $|1S^2|1P^6|1D^{10}|1F^2|$. Finally, we conclude that each coordinative bond contains 7/8ths bonding orbitals and 1/8 nonbonding orbitals by applying bond order analysis.

## Results and discussion

**The similarity of $[Ba(CO)_8]^{2-}$ and $[BaBe_8]^{2-}$.** Traditionally, Jellium model is generally applied to explain the stability and electronic structure of metal cluster[14,15], such as the magic stability of the icosahedral $Al_{13}^-$ cluster[16–18]. The metal cluster can be viewed as a superatom, which is comprised from positive charge of atomic nuclei and the innermost electrons, where the valence electrons are subjected to an external potential in the delocalized motion as $|1S^2|1P^6|1D^{10}2S^2|1F^{14}2P^6|…$, where the resulting magic numbers are 2, 8, 20, 40, …. (To distinguish the electronic shells of atoms, the super shells are depicted as capital letters.)

Therefore, we hypothesize a metal cluster containing eight coordinative bonds and two d → π* backdonation bonds, such as $[BaBe_8]^{2-}$, to compare with $[Ba(CO)_8]^{2-}$ analogically. Both of the anions strictly satisfy the closed-shell "20e-superatoms" with $O_h$ symmetry.

As for the analogical octa-coordinative complex, the $O_h$-symmetric $[BaBe_8]^{2-}$ are optimized at M06-2×/def2tzvpp level of theory[19,20] in Gaussian 09[21] to obtain the singlet electronic ground state. The valence electron configuration of $[BaBe_8]^{2-}$ is $a_{1g}^2t_{1u}^6t_{2g}^6a_{2u}^2e_g^4$ with the HOMO-LUMO gap of 2.77 eV. Based on the Jellium model, the electrons fulfill the ten valence orbitals following the configuration of $|1S^2|1P^6|1D^6|1F^2|1D^4|$, where D orbitals are split into two groups as ($1D_{xy}$, $1D_{yz}$ and $1D_{xz}$) and ($1D_{x^2−y^2}$ and $1D_{z^2}$). It should be noticed that the $a_{2u}$ orbital is a 1F orbital rather than the 2S orbital, where the energy level of 1F orbital is in the middle of the two groups of 1D orbitals.

Based on the calculation, $a_{2u}$ orbital with f symmetry is shown in Fig. 1. Furthermore, there is a cubic field in all the $M^q(CO)_8$ complexes or ions, which affect the seven-degenerate 1F orbitals in the coordination, in which the $a_{2u}$-symmetric $1F_{xyz}$ orbital matches the orbital symmetry in the cubic field of coordination. Therefore, the energy of $1F_{xyz}$ orbital is lower than the 2S orbital caused by the splitting of F orbitals (Supplementary Fig. 1). The sequences of energy levels of the orbitals are the results of the splitting of 1D and 1F orbitals. In the configuration, the number of valence electrons just equals the magic number "20" achieving magical stability.

The comparison of the molecular orbitals of $[Ba(CO)_8]^{2-}$ and $[Ba(Be)_8]^{2-}$ is shown in Fig. 1. Although $[Ba(CO)_8]^{2-}$ is not a metal cluster, its configuration can be similarly described as $|1S^2|1P^6|1D^6|1F^2|1D^4|$ based on the Jellium model. Furthermore, $a_{1g}$, $t_{1u}$ and $t_{2g}$ orbitals of $[Ba(CO)_8]^{2-}$ correspond to the 1S, 1P orbitals and $1D_{xy}/1D_{yz}/1D_{xz}$ super orbitals. The $e_g$ MOs correspond to the $1D_{x^2−y^2}$ and $1D_{z^2}$ orbitals as two π-type orbitals, which are the two d → π* bonds donated from 5d AOs of Ba ($5d_{x^2−y^2}$ and $5d_{z^2}$ AOs) to the 2p AOs of eight CO groups. The ligand-only $a_{2u}$ orbital composed of eight carbonyl groups is also defined as the 1F super orbital based on the diagram in Fig. 1. The similarity of $[Ba(CO)_8]^{2-}$ and $[BaBe_8]^{2-}$ indicates we can understand the valence orbitals of $[Ba(CO)_8]^{2-}$ as a "20e-superatom" based on the Jellium model.

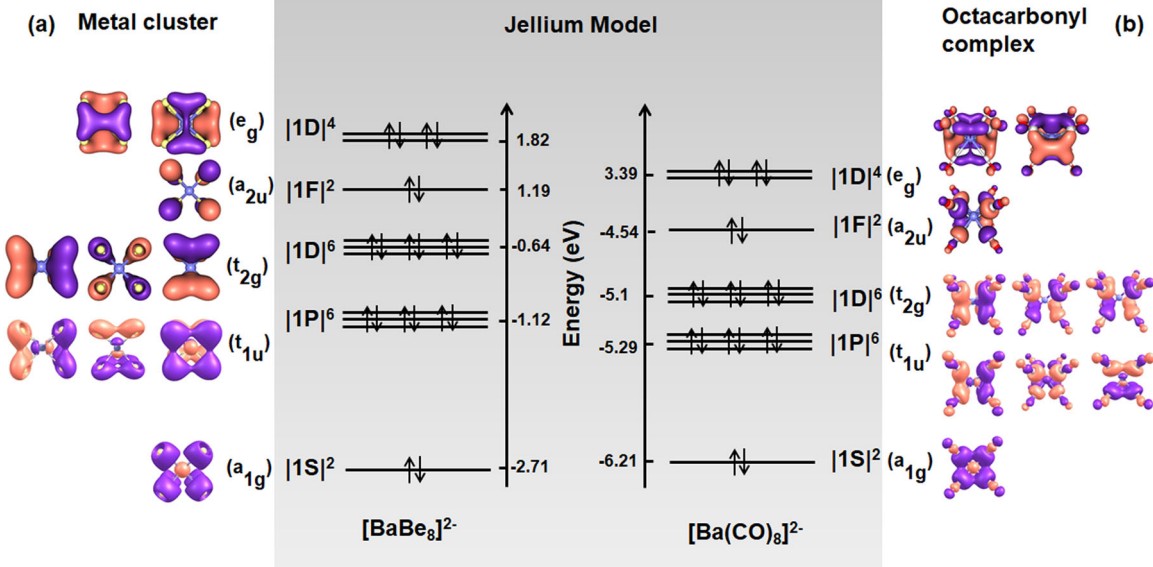

**Fig. 1 The molecular orbitals listed on the basis of the Jellium model. a** The molecular orbitals of the singlet $[Ba(Be)_8]^{2-}$. **b** The molecular orbitals of the singlet $[Ba(CO)_8]^{2-}$.

For another aspect, we apply MO theory to understand the eight σ-type coordinative bonds in $[Ba(CO)_8]^{2-}$. There should be 15 MOs contributed by 7 valence AOs of Ba atom ($d^3sp^3$) and 8 AOs of LPs donated from 8 carbonyl groups in forming eight σ (Ba-C) bonds based on the MO theory. Therefore, the 15 MOs should be composed of 7 bonding, 1 nonbonding, and 7 antibonding orbitals fulfilled by 16 electrons successively. In the 15 MOs, the 7 bonding orbitals correspond to the $a_{1g}$, $t_{1u}$ and $t_{2g}$ orbitals. The nonbonding orbital exactly corresponds to the ligand-only $a_{2u}$ orbital. This is also consistent with the sequence of molecular orbital level energy, where the energy of $1F_{xyz}$ orbital as the nonbonding orbital is higher than that of ($1D_{xy, xz, yz}$) and lower than the energy of $1D_{x^2-y^2}$ and $1D_{z^2}$ orbitals.

**The 20-electron model for $[M(CO)_8]^q$ (M, q = Ca, 2−; Sc, 1−; Ti, 0; V, 1+; Cr, 2+).** To reveal the mystery of the state of the ligand-only orbital ($a_{2u}$ orbital), we propose to explain the bonding structures of $[M(CO)_8]^q$ (M, q = Ca, 2−; Sc, 1−; Ti, 0; V, 1+; Cr, 2+) as the singlet "20e-superatom" models based on the comparison in Fig. 1. The typical molecules selected as $[Ca(CO)_8]^{2-}$, $[Sc(CO)_8]^-$, $Ti(CO)_8$, $[V(CO)_8]^+$ and $[Cr(CO)_8]^{2+}$ with $O_h$ symmetry are optimized under M06-2×/def2tzvpp theoretical level to obtain their singlet electronic ground states. All the structures with the coordination information are listed in the Supplementary Information (Supplementary Figs. 2−12 and Supplementary Tables 4−14). All of them have the same valence electron configuration of $a_{1g}^2t_{1u}^6t_{2g}^6a_{2u}^2e_g^4$. Each orbital contributed by both ligand and center metal can be directly demonstrated by manipulation with structural subunits of Metal-C bonds, which is confirmed by natural bonding orbital (NBO) analysis by using the adaptive natural density partitioning (AdNDP) method[22]. The AdNDP analyses are shown in Fig. 2a.

As for $Ti(CO)_8$ (Fig. 2a), the HOMO-LUMO gap of $Ti(CO)_8$ is 5.47 eV under the same theoretical level of M062x/def2tzvpp. As for $[Ca(CO)_8]^{2-}$, $[Sc(CO)_8]^-$, $[V(CO)_8]^+$ and $[Cr(CO)_8]^{2+}$, the HOMO-LUMO gaps are 3.12, 4.28, 6.94 and 9.12 eV, respectively. There are no virtual frequencies in all the five $[M(CO)_8]^q$ (M, q = Ca, 2−; Sc, 1−; Ti, 0; V, 1+; Cr, 2+) complex or ions. The vibrational frequencies of five $M^q(CO)_8$ are listed in the Supplementary Information (Supplementary Table 1). Because

we do not find the stable conformation of $[Ca(CO)_7]^{2-}$, we only calculate the corresponding dissociation energies of $M^q(CO)_8 \rightarrow M^q(CO)_7 + CO$ for $[Sc(CO)_8]^-$, $[Ti(CO)_8]$, $[V(CO)_8]^+$ and $[Cr(CO)_8]^{2+}$. (The results are listed in Supplementary Table 2.) All the results indicate that $M^q(CO)_8$ complex/ions are the minimums on the PESs. All the O atoms and carbonyl ligands appear as the same configurations, which include 8 LPs of electrons localized on O atoms, 8 $\sigma_{CO}$ and 16 $\pi_{CO}$ bonds between C and O atoms. On the basis of the AdNDP results[22], the occupancy numbers ($|ONs|$) of the orbitals of the ligands (in Fig. 2a) are very close to the ideal value $2.0|e|$, suggesting the reasonable valence properties.

We accomplish the similar MO analysis of $Ti(CO)_8$ with that of $[Ba(CO)_8]^{2-}$ to obtain the similar results. $a_{1g}$ and $t_{1u}$ orbitals correspond to the 1S and 1P orbitals. $t_{2g}$ orbitals correspond to the $1D_{xy}$, $1D_{yz}$ and $1D_{xz}$ orbitals. $e_g$ orbitals correspond to the $1D_{x^2-y^2}$ and $1D_{z^2}$ orbitals, which are the two $d \rightarrow \pi^*$ bonds donated from $3d_{x^2-y^2}$ and $3d_{z^2}$ AOs of Ti to the antibonding $\pi^*$ MOs of carbonyl groups. The ligand-only $a_{2u}$ orbital corresponds to the 1 F orbital. On the basis of the Jellium model with the configuration of $|1S^2|1P^6|1D^{10}|1F^2|$, the $a_{2u}$ orbital is not isolated but as a component in forming the eight coordinative bonds, which can be further confirmed by the AdNDP analyses for the coordinative bonds of $Ti(CO)_8$. Each Ti-C coordinative bond can be viewed as a 9-center-2-electron bond (9c-2e). The AdNDP analysis reveals seven 9c-2e bonds (including one 1S, 3-degenerate 1P, 3-degenerate 1D and one 1F super orbitals) correspond to the eight σ bonds. The other two π-backdonation bonds ($1D_{x^2-y^2}$ and $1D_{z^2}$ super orbitals) between titanium and the eight ligands can be viewed as 17c-2e bonds. All the $|ONs|$ are closed to the ideal value of $2.0|e|$ in Table 1. We deal with the other four $[M(CO)_8]^q$ ions (M, q = Ca, 2−; Sc, 1−; V, 1+; Cr, 2+) in the exactly same pathway to obtain the same configuration of $|1S^2|1P^6|1D^{10}|1F^2|$ based on the Jellium model and similar results based on the AdNDP method.

As for the $a_{2u}$ orbital in $Ti(CO)_8$, it is a special 9c-2e bond in the molecular orbitals with the $|ONs|$ of 1.98, which is contributed by the ligands entirely because all the nine valence AOs of Ti are composed of the nine MOs including seven σ-orbitals and two $d \rightarrow \pi^*$ orbitals.

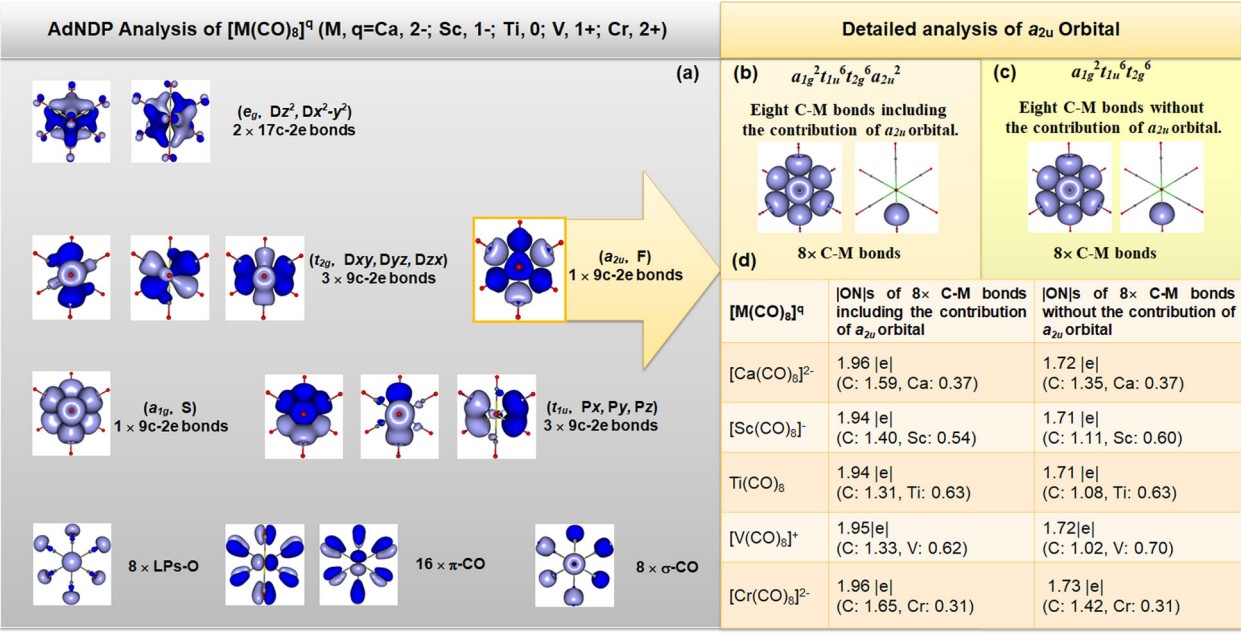

**Fig. 2 AdNDP analyses of metal−ligand bonds. a** AdNDP analyses of $[M(CO)_8]^q$ (M, q = Ca, 2−; Sc, 1−; Ti, 0; V, 1+; Cr, 2+) complex/ions. **b** The detailed AdNDP analyses of (|ONs|) of the five $M^q(CO)_8$ complex/ions with the participation of $a_{2u}$ orbital. **c** The detailed AdNDP analyses of (|ONs|) of the five $M^q(CO)_8$ complex/ions without the participation of $a_{2u}$ orbital. **d** The occupied numbers (|ONs|) of $M^q(CO)_8$ complex/ions with/without the participation of $a_{2u}$ orbital.

**Table 1 The results of AdNDP analysis and occupied numbers of $[M(CO)_8]^q$ (M, q = Ca, 2−; Sc, 1−; Ti, 0; V, 1+; Cr, 2+).**

| AdNDP analysis | | Orbital type | Occupied numbers (|ONs|) of orbitals | | | | |
|---|---|---|---|---|---|---|---|
| | | | $[Ca(CO)_8]^{2-}$ | $[Sc(CO)_8]^{-}$ | $Ti(CO)_8$ | $[V(CO)_8]^{+}$ | $[Cr(CO)_8]^{2-}$ |
| Localized MOs of CO | 8 × LP(O) | LPs on O atom | 1.98\|e\| | 1.98\|e\| | 1.97\|e\| | 2.00\|e\| | 1.98\|e\| |
| | 16 × π(CO) | π-type bonds of CO | 2.00\|e\| (C: 0.47, O: 1.53) | 1.99\|e\| (C: 0.49, O: 1.50) | 2.00\|e\| (C: 0.52, O: 1.48) | 2.00\|e\| (C: 0.59, O: 1.41) | 2.00\|e\| (C: 0.47, O: 1.53) |
| | 8 × σ(CO) | σ-type bond of CO | 2.00\|e\| (C: 0.56, O: 1.44) | 2.00\|e\| (C: 0.57, O: 1.43) | 2.00\|e\| (C: 0.58, O: 1.42) | 2.00\|e\| (C: 0.54, O: 1.46) | 2.00\|e\| (C: 0.58, O: 1.42) |
| Delocalized MOs among ligands and metal | 1 × 9c-2e bonds | S-type bond with $a_{1g}$, symmetry | 2.00\|e\| (C: 1.61, Ca: 0.39 | 2.00\|e\| (C: 1.54, Sc: 0.43 | 2.00\|e\| (C: 1.55, Ti: 0.45) | 2.00\|e\| (C: 1.54, V: 0.46) | 1.99\|e\| (C: 1.46, Cr: 0.53) |
| | 3 × 9c-2e bonds | Px, Py and Pz bonds with $t_{1u}$, symmetry | 1.99\|e\| (C: 1.63, Ca: 0.36) | 1.98\|e\| (C: 1.20, Sc: 0.78) | 1.97\|e\| (C: 1.31, Ti: 0.46) | 1.98\|e\| (C: 1.08, V: 0.90) | 1.99\|e\| (C: 1.60, Cr: 0.39) |
| | 3 × 9c-2e bonds | Dxy, Dyz, Dzx bonds with $t_{2g}$, symmetry | 1.99\|e\| (C: 1.51, Ca: 0.48) | 1.98\|e\| (C: 1.56, Sc: 0.42) | 1.98\|e\| (C: 1.07, Ti: 0.91) | 1.98\|e\| (C: 1.53, V: 0.45) | 1.98\|e\| (C: 1.75, Cr: 0.23) |
| | 1 × 9c-2e bonds | F bond with $a_{2u}$, symmetry | 1.98\|e\| (C: 1.98, Ca: 0) | 1.98\|e\| (C: 1.98, Sc: 0) | 1.97\|e\|) (C: 1.97, Ti: 0) | 1.98\|e\|) (C: 1.98, V: 0) | 1.99\|e\|) (C: 1.99, Cr: 0) |
| | 2 × 17c-2e bonds | Dz², Dx²−y² bonds with $e_g$, symmetry | 2.00\|e\| (C: 1.12, O: 0.35, Ca: 0.53) | 1.98\|e\| (C: 0.85, O: 0.29, Sc: 0.86) | 2.00\|e\| (C: 0.58, O: 0.22, Ti: 1.20) | 2.00\|e\| (C: 0.32, O: 0.12, V: 1.56) | 2.00\|e\| (C: 0.17, O: 0.06, Cr: 1.77) |

**The contribution of "ligand-only" $a_{2u}$ orbital**. In order to make clear whether the $a_{2u}$ orbital in any $[M(CO)_8]^q$ (M, q = Ca, 2−; Sc, 1−; Ti, 0; V, 1+; Cr, 2+) participates in the formation of eight coordinative bonds, we apply the AdNDP method to compare the |ONs| of eight Ti-C orbitals of $Ti(CO)_8$ with and without participation of the $a_{2u}$ orbital (Fig. 2b, c). The |ONs| are 1.94|e| with the contribution of $a_{2u}$ orbital if we localize eight orbitals

$(a_{1g} + 3t_{1u} + 3t_{2g} + a_{2u})$ in forming eight coordinative bonds, where C contributes 1.31|e| and Ti contributes 0.63|e| in each covalence bond. However, the |ONs| are decreased to 1.71|e| when we localize only $a_{1g}$, $3t_{1u}$ and $3t_{2g}$ orbitals in forming the eight coordinative orbitals, where the component of C is decreased from 1.31|e| to 1.08|e|. However, the |ONs| of Ti are still 0.63|e|, which equals to the value including the contribution

of $a_{2u}$ orbital. This means the reduced |ONs| are caused by the deduction of the $a_{2u}$ orbital contributed by the ligands, which demonstrates the $a_{2u}$ orbital is still participating in the coordination, although it is contributed only by the ligands.

For another aspect, the diminution of |ONs| from 1.94|e| to 1.71|e| suggests around 1/8 components (Table 2) of the coordinative orbitals have been deducted, which is the contribution of the $a_{2u}$ orbital in the coordination. After dealing with $a_{2u}$ orbitals of the other four $M^q(CO)_8$ ions ($M^q = Ca^{2-}$, $Sc^-$, $V^+$ or $Cr^{2+}$) (Fig. 2d) with the same methods, we also obtain similar results. The diminution of |ONs| from 1.96 (1.94/1.95/1.96)

to 1.72 (1.71/1.72/1.73) of $[Ca(CO)_8]^{2-}$ ($[Sc(CO)_8]^-$/$[V(CO)_8]^+$/$[Cr(CO)_8]^{2+}$) suggests the $a_{2u}$ orbital contribute around 1/8 components in the coordinative orbital of the five $M^q(CO)_8$ systems (Table 2).

In order to make sure the ratio is 1:7 between bonding orbitals and nonbonding orbitals, we further compare the bond population of Metal-C bond of "20-e" $Ti(CO)_8$ ($O_h$ symmetry) with the standard "18-e" $Ti(CO)_7$ ($C_{3V}$ symmetry), based on the results of Wiberg bond index (WBI)[23] and Mayer bond order (MBO)[24] calculations under the same theoretical level in Table 3. There are exactly seven bonding orbitals in $Ti(CO)_7$ with the bond order of 7 because there is no $a_{2u}$ orbital, which is different from $Ti(CO)_8$ with the bond order of 7 but including 7 bonding orbitals and 1 nonbonding orbital. So the differences of the population are caused by the "ligand-only" $a_{2u}$ orbital. The ratio of the Ti-C bond populations between $Ti(CO)_8$ and $Ti(CO)_7$ is the ratio of the bonding orbitals in $Ti(CO)_8$. Therefore, it can be concluded that the bonding orbitals occupy around 7/8ths (nonbonding orbital contribute 1/8 components) in the coordination. The similar comparisons for the other three $[M(CO)_8]^q$ ions (M, q = Sc, 1−; V, 1+; Cr, 2+) are listed in the Supplementary Information (Supplementary Table 3), where we can obtain similar conclusions.

So in the octacarbonyl metal complex or ions $[M(CO)_8]^q$ (M, q = Ca, 2−; Sc, 1−; Ti, 0; V, 1+; Cr, 2+; Ba, 2−), it is more reasonable to view the ligand-only $a_{2u}$ orbital as a nonbonding orbital contributes to form the eight coordinative orbitals ($a_{1g}^2 t_{1u}^6 t_{2g}^6 a_{2u}^2$), where each coordinative bond contains 1/8 nonbonding orbitals and 7/8 bonding orbitals as shown in Fig. 3. Based on the MO theory, there are seven bonding orbitals corresponding to the eight 7/8ths bonding orbitals, one nonbonding orbital corresponding to the eight 1/8th nonbonding orbitals in forming the eight coordinative bonds. The eight coordinative bonds with the bond order of 7 is consistent with the

**Table 2 The contribution of the ligand-only $a_{2u}$ orbital in the coordinative bond.**

|  | $[Ca(CO)_8]^{2-}$ | $[Sc(CO)_8]^-$ | $Ti(CO)_8$ | $[V(CO)_8]^+$ | $[Cr(CO)_8]^{2+}$ |
|---|---|---|---|---|---|
| |ONs|1 | 1.96 | 1.94 | 1.94 | 1.95 | 1.96 |
| |ONs|2 | 1.72 | 1.71 | 1.71 | 1.72 | 1.73 |
| Ratio | 12.2% | 11.9% | 11.8% | 11.8% | 11.7% |

|ONs|1 is the occupation number including the contribution of $a_{2u}$ orbital. |ONs|2 is the occupation number without the contribution of $a_{2u}$ orbital.
Ratio is the components of the $a_{2u}$ orbital, which is calculated as: (|ONs|1 − |ONs|2)/|ONs|1.

**Table 3 The ratio of bonding orbitals in $Ti(CO)_8$ by population analysis of $Ti(CO)_7$ and $Ti(CO)_8$.**

|  | $Ti(CO)_8$ | $Ti(CO)_7$ | Ratio |
|---|---|---|---|
| WBI(Ti-C) | 0.88 | 1.03 | 0.854/1 |
| MBO(Ti-C) | 0.76 | 0.91 | 0.835/1 |

*The ratio of the bonding orbital in $Ti(CO)_8$.

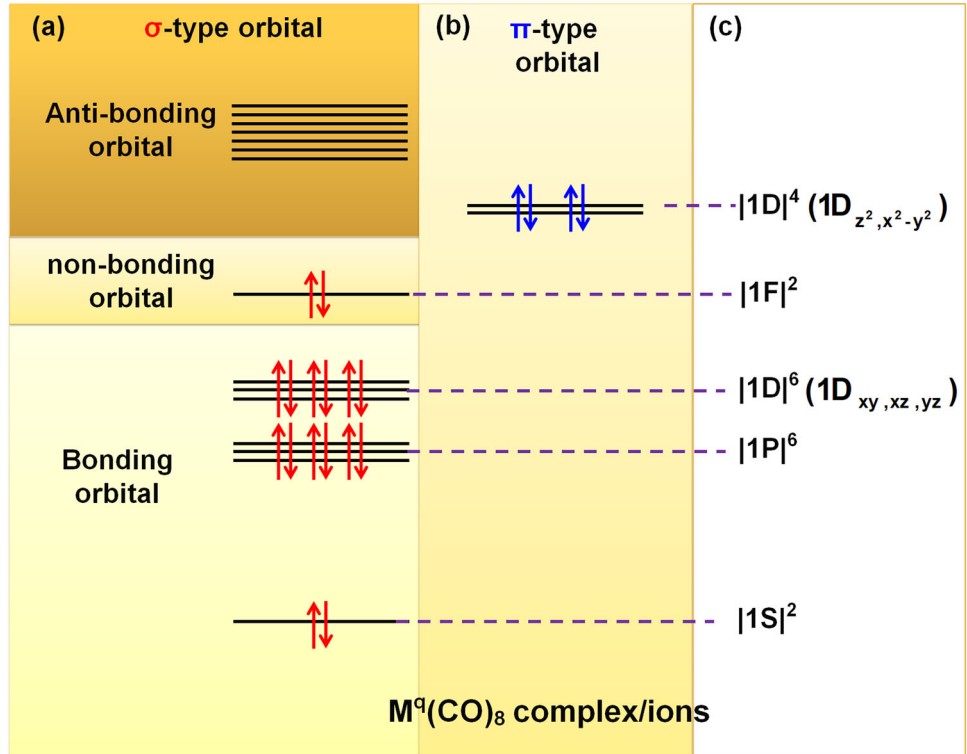

**Fig. 3 MO diagram of $[M(CO)_8]^q$ (M, q = Ca, 2−; Sc, 1−; Ti, 0; V, 1+; Cr, 2+; Ba, 2−). a** The σ orbital diagram of $[M(CO)_8]^q$. **b** The d−π* backdonation bonds of $[M(CO)_8]^q$. **c** The configuration as Jellium model by viewing $M^q(CO)_8$ as a "20-e superatom".

results of both VB and HO theories. Furthermore, the coordinative structures of the octacarbonyl metal complexes or ions can be viewed as a "20-e superatom" with the configuration of $|1S^2|1P^6|1D^{10}|1\ F^2|$ based on the unambiguous Jellium model. In this model, $|1D^{10}|$ orbitals split into two groups including the three σ-type coordinative orbitals ($|1D^6|$) with lower energy and two π-type backdonation orbitals ($|1D^4|$) with higher energy. The nonbonding orbital with $a_{2u}$ symmetry corresponds to $|1F|$ super orbital ($1F_{xyz}$) rather than the $|2S|$ orbital, which is caused by the splitting of 1F orbitals (Fig. 3c).

This research demonstrates the Jellium model is a powerful analysis tool that can not only be applied on metal clusters but also be useful to understand the high symmetric octacarbonyl metal complexes or ions. The $a_{2u}$ orbital is indeed a ligand-only orbital but contributes 1/8 components in each coordinative bonds in $[M(CO)_8]^q$ complex or ions $[M(CO)_8]^q$ (M, q = Ca, 2−; Sc, 1−; Ti, 0; V, 1+; Cr, 2+; Ba, 2−). Therefore, 20-e model with the configuration $|1S^2|1P^6|1D^{10}|1F^2|$ is an appropriate description of the valence bonding structures of $M(CO)_8$ or $TM(CO)_8$ complexes and ions.

## Methods

**Optimization**. On the basis of the synthesized octacarbonyl alkali earth metal complexes, we have designed six $[M(CO)_8]^q$ complex or ions (M, q = Ca, 2−; Sc, 1−; Ti, 0; V, 1+; Cr, 2+; Ba, 2−) and a metal cluster $[BaBe_8]^{2-}$. All the structures are optimized at M06-2×/def2tzvpp level of theory[19,20] in Gaussian 09 (Version E. 01)[21] to obtain their singlet electronic ground states. All the structures with the coordination information are listed in the Supplementary Information (Supplementary Figs. 2–12 and Supplementary Tables 4–14).

**Analysis of structural stability**. Based on the optimized structures, we calculated the vibrational frequencies (cm$^{-1}$) and HOMO-LUMO gaps (eV) under M06-2×/def2tzvpp theoretical level. Because we did not find the stable conformation of $[Ca(CO)_7]^{2-}$, we only calculated the corresponding dissociation energies of $[M(CO)_8]^q \rightarrow [M(CO)_7]^q + CO$ for $[Sc(CO)_8]^-$, $[Ti(CO)_8]$, $[V(CO)_8]^+$ and $[Cr(CO)_8]^{2+}$. All the structures are optimized under the same theoretical methods. The dissociation energy is the Gibbs free energy variation of the corresponding reaction. All the results are listed in Supplementary Tables 1 and 2.

**Analysis of MOs**. With the same methods, each orbital contributed by both ligand and center metal has been manipulated with structural subunits of Metal-C bonds, which is confirmed by the NBO analysis by using the AdNDP method[22]. MO visualization is performed using MOLEKEL 5.4 software[25]. On the basis of optimized structures, the WBI has been obtained directly from Gaussian 09 program by using NBO population analysis. The MBO is obtained by using Multiwfn tools (Version 3.6)[26].

## Data availability

The data that support the findings of this study are available from the corresponding authors upon reasonable request.

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

## Acknowledgements

This work is financed by the National Natural Science Foundation of China (21701001, 21873001) and by the Foundation of Distinguished Young Scientists of Anhui Province. The calculations are carried out at the High-Performance Computing Center of Anhui University.

## Author contributions

K.W. performed the analysis and wrote the manuscript. K.W. and L.C. designed the models and conceived the project. K.W., C.X, D.L. and L.C. performed the structure optimization and AdNDP analysis. L.C. supervised the project.

## Competing interests

The authors declare no competing interests.
