## [Peer Review File · Communications Chemistry]

Reviewers' comments:

Reviewer #1 (Remarks to the Author):

This manuscript reports a theoretical study on the valence bond structure of the recently reported octacarbonyl complexes. The authors try to use the Jellium model to understand the electronic structure of octacarbonyl complexes. My main criticism concerns the fact that the authors claim the 20-e model Jellium model is the most appropriate explanation in understanding the valence bonding structure of $M(\text{CO})_8$ complexes. As already discussed in ref. 3 and 8, the bonding structure of these octacarbonyl complexes are well described using the molecular orbital theory for such small molecule systems. These complexes are 20e species with 18 effective valence electrons on the metal center, and thus fulfill the 18 electron rule. However, the effort to use Jellium model to interpret the valence bonding structure qualitatively is interesting. The manuscript is not well organized, and there are a number of significant points to be clarified. Therefore, a thoroughly revised manuscript that addresses all these issues (see below) might be acceptable for publication:

1. The title is confusion and imprecision. " $M(\text{CO})_8$ " should be " $M_q(\text{CO})_8$ ($M_q = \text{Ca}^{2-}, \text{Sc}^-, \text{Ti}, \text{Cr}^{2+}$)".
2. The statement "the HOMO-LUMO gaps are 3.12 eV and 9.12 eV, both of which indicate they are stable compounds" is incorrect. To judge the stability of a compound, the authors should confirm that the optimized structure is a minimum on the PES and that it is stable with respect to dissociation $M(\text{CO})_8 \rightarrow M(\text{CO})_7 + \text{CO}$. In this regard, the $\text{Ti}(\text{CO})_8$ and $\text{Cr}(\text{CO})_8^{2+}$ complexes are likely not stable species. The authors should add the calculated vibrational frequencies and the CO dissociation energies in SI.
3. For completeness, the authors should include $\text{Sc}(\text{CO})_8^-$ and $\text{V}(\text{CO})_8^+$ as well.
4. Regarding the introduction, several "metals" should be "transition metals", as the 18e rule is generally applicable only for transition metals.
5. Figure 2 is very crowded and difficult to follow. The authors should clean the figure and put the numbers into a table.

Reviewer #2 (Remarks to the Author):

The authors have analyzed the bonding of the recently introduced octacarbonyl metal complexes $M(\text{CO})_8$. Based on the Jellium model, the stability of these complexes is attributed to satisfying the Jellium model with 20 electrons. In this way, each coordinative bond contains 7/8 of bonding and 1/8 of non-bonding contributions. This non-bonding contribution comes from the a_{2u} orbital, which is extensively discussed in this work. Overall, this manuscript explains the presence of 8 coordinative bonds with a bond order of 7, supporting both valence bond and hybrid bond theories. Its interest and impact to the community, makes me recommend its publication in Communications Chemistry. However, before its acceptance, several points should be either addressed or clarified:

- 1) $M(\text{CO})_8$ with $M = \text{Ca}, \text{Sr}, \text{or Ba}$) were initially reported to be neutral species, thus with 18 electrons. However, in this manuscript, the authors first refer to the neutral species, however when referring to the Jellium model, the dianion species are taken into analysis, thus with 20 electrons. The authors should clearly specify the reason why they go from the initial neutral species, with O_h symmetry and a triplet electronic ground state, to the dianions, with O_h symmetry as well, but closed shell singlet state.
- 2) Regarding the Jellium model, why is 2S not occupied? Is it higher in energy than 1F(2)?
- 3) Very recently, Poater and Solà have extended the Jellium model to open-shell half-filled systems (Chem. Comm. 2019, 55, 5559-5562). Thus, and related to point 1 above, could this new rule be applied to neutral $M(\text{CO})_8$ systems with triplet ground state? The two highest eg (1D(2)) orbitals are half-filled, thus fulfilling this new rule. This would also justify the stability of these

systems, not being necessary to go into the dianions, and to further support the theory presented in this manuscript. Both sets of systems could be enclosed and discussed, both of them supported by Jellium model, the one with 20 electrons with the closed-shell model, whereas that with 18 electrons with the open-shell model.

4) BaBe8 is first presented as neutral (title), but the whole discussion is based on the dianion. Please correct accordingly.

5) If all calculations have been performed with the M06-2X functional, why for Ti(CO)₈ BP86 is used? (second paragraph in page 3) Maybe this is an error, as in the Supporting Information only M06-2X has been used.

6) In the caption of Figure 2, please add also the colors used for each of the three systems: red, black and blue.

7) Can the authors justify the reason why 1D(4) orbitals can be either lower (Figure 3) or higher in energy than 1F(2) (Figure 1)? How does the change of M affect these trends?

8) Finally, several misprints along the manuscript should be corrected.

Reviewer #3 (Remarks to the Author):

Comments:

In this paper, the authors chose a very interesting topic. The bonding nature of M(CO)₈ (M = Ca, Sr, or Ba) has been thoroughly discussed in previous reports [Science 361, 912-916 (2018); Science 365, DOI: 10.1126/science.aay5021 (2019); Science 365, DOI: 10.1126/science.aay2355 (2019)]. It is innovative for the authors to use the jellium model to explain the valence bonding structure of M(CO)₈ (M = Ca, Sr, or Ba) complexes. However, I think this classical jellium model is only adequate to metallic clusters. For example, the t_{2g} orbitals of M(CO)₈ in Fig. 1 are assigned to 1D superatomic orbitals are not reasonable since they are more like F orbitals in the jellium model. The same cases are also found in Fig. 2a, wherein the t_{1u} are considered as 1P orbitals are ridiculous. Hence, I think this paper is not appropriate to be published in this journal. If the authors want to publish this work in another theoretical journal, the following mistakes should be seriously modified:

On Pg 2, line 13(left column) "alkali metal" should be "alkaline earth metal"; on Pg 3, line 2 of the second paragraph, the computational method "BP86/def2tzvpp" should be "M06-2X/def2tzvpp" according to their supporting information. Thus, the authors should check the main text carefully again.

Response to the Reviewers

Reviewers' comments:

Reviewer #1 (Remarks to the Author):

This manuscript reports a theoretical study on the valence bond structure of the recently reported octacarbonyl complexes. The authors try to use the Jellium model to understand the electronic structure of octacarbonyl complexes. My main criticism concerns the fact that the authors claim the 20-e model Jellium model is the most appropriate explanation in understanding the valence bonding structure of $M(\text{CO})_8$ complexes. As already discussed in ref. 3 and 8, the bonding structure of these octacarbonyl complexes are well described using the molecular orbital theory for such small molecule systems. These complexes are 20e species with 18 effective valence electrons on the metal center, and thus fulfill the 18 electron rule. However, the effort to use Jellium model to interpret the valence bonding structure qualitatively is interesting. The manuscript is not well organized, and there are a number of significant points to be clarified. Therefore, a thoroughly revised manuscript that addresses all these issues (see below) might be acceptable for publication:

1. The title is confusion and imprecision. " $M(\text{CO})_8$ " should be " $M^q(\text{CO})_8$ ($M^q=\text{Ca}^{2-}, \text{Sc}^-, \text{Ti}, \text{Cr}^{2+}$)".

We have modified as the suggestion. The title is "Valence bonding structures of $M^q(\text{CO})_8$ complex/ions ($M^q=\text{Ca}^{2-}, \text{Sc}^-, \text{Ti}, \text{V}^+, \text{Cr}^{2+}$ or Ba^{2-}): 18-electron or 20-electron?"

We also modified the relevant descriptions in the revised manuscript:

1. By comparing $[\text{Ba}(\text{CO})_8]^{2-}$ and metal clusters of $[\text{BaBe}_8]^{2-}$ analogically, we demonstrate Jellium model is also a useful tool in understanding the electronic structures of $M^q(\text{CO})_8$ ($M^q=\text{Ca}^{2-}, \text{Sc}^-, \text{Ti}, \text{V}^+, \text{Cr}^{2+}$ or Ba^{2-}) rather than only for metal clusters. By applying Jellium model, the magic stabilities of such $M^q(\text{CO})_8$ complex or ions are because the structure is satisfying the '20-electron' rule.
2. Therefore, we design a series of O_h -symmetric and singlet-state $M^q(\text{CO})_8$ ($M^q=\text{Ca}^{2-}, \text{Sc}^-, \text{Ti}, \text{V}^+, \text{Cr}^{2+}$ or Ba^{2-}) complex/ions theoretically to learn their bonding structures. Such $M^q(\text{CO})_8$ complex/ions are 20-e closed-shell molecules based on the configuration³, which is exactly satisfying the magic stability of Jellium model
3. Furthermore, there is a cubic field in all the $M^q(\text{CO})_8$ complexes or ions, which affect the 7-degenerate 1F orbitals in the coordination, in which the a_{2u} -symmetric $1F_{xyz}$ orbital matches the orbital symmetry in the cubic field of coordination.
4. To reveal the mystery of the state of the ligand-only orbital (a_{2u} orbital), we propose to explain the bonding structures of $M^q(\text{CO})_8$ ($M^q=\text{Ca}^{2-}, \text{Sc}^-, \text{Ti}, \text{V}^+$ or Cr^{2+}) as the singlet '20e-superatom' models based on the comparison in Fig. 1.
5. There are no virtual frequencies in all the five $M^q(\text{CO})_8$ ($M^q=\text{Ca}^{2-}, \text{Sc}^-, \text{Ti}, \text{V}^+$ or Cr^{2+}) complex or ions. The vibrational frequencies of five $M^q(\text{CO})_8$ are listed in the Supporting information (SI-Table S1). Because we do not find the stable conformation of $[\text{Ca}(\text{CO})_7]^{2-}$, we only calculate the corresponding dissociation energies of $M^q(\text{CO})_8 \rightarrow M^q(\text{CO})_7 + \text{CO}$ for $[\text{Sc}(\text{CO})_8]^-$, $[\text{Ti}(\text{CO})_8]$, $[\text{V}(\text{CO})_8]^+$ and $[\text{Cr}(\text{CO})_8]^{2+}$ (The results are listed in SI-Table S2).
6. We deal with the other four $M^q(\text{CO})_8$ ions ($M^q=\text{Ca}^{2-}, \text{Sc}^-, \text{V}^+$ or Cr^{2+}) in the exactly same pathway to obtain the same configuration of $|1S^2|1P^6|1D^{10}|1F^2|$ based on the jellium model and similar results from AdNDP method.
7. In order to make clear whether the a_{2u} orbital in any $M^q(\text{CO})_8$ ($M^q=\text{Ca}^{2-}, \text{Sc}^-, \text{Ti}, \text{V}^+$ or Cr^{2+}) participates in the formation of eight coordinative bonds...
8. After dealing with a_{2u} orbitals of the other four $M^q(\text{CO})_8$ ions ($M^q=\text{Ca}^{2-}, \text{Sc}^-, \text{V}^+$ or Cr^{2+}) (Fig. 2d) with the same methods, we also obtain the similar results.
9. The similar comparison for the other three $M^q(\text{CO})_8$ ions ($M^q=\text{Sc}^-, \text{V}^+$ or Cr^{2+}) are listed in the supporting information (SI-table S3), where we can obtain the similar conclusions.

10. The a_{2u} orbital is indeed a ligand-only orbital but contributing as 1/8 components in each coordinative bonds in $M^q(\text{CO})_8$ complex or ions ($M^q = \text{Ca}^{2+}, \text{Sc}, \text{Ti}, \text{V}^+, \text{Cr}^{2+}$ or Ba^{2+}).

2. The statement “the HOMO-LUMO gaps are 3.12 eV and 9.12 eV, both of which indicate they are stable compounds” is incorrect. To judge the stability of a compound, the authors should confirm that the optimized structure is a minimum on the PES and that it is stable with respect to dissociation $M(\text{CO})_8 \rightarrow M(\text{CO})_7 + \text{CO}$. In this regard, the $\text{Ti}(\text{CO})_8$ and $\text{Cr}(\text{CO})_8^{2+}$ complexes are likely not stable species. The authors should add the calculated vibrational frequencies and the CO dissociation energies in SI.

We have added the corresponding energies in SI-Table S1. As for the dissociation $M(\text{CO})_8 \rightarrow M(\text{CO})_7 + \text{CO}$, we do not find the stable conformation of $[\text{Ca}(\text{CO})_7]^{2+}$. But there are no virtual frequencies in the results of $[\text{Ca}(\text{CO})_8]^{2+}$, indicating $[\text{Ca}(\text{CO})_8]^{2+}$ is a stable structure on the PES. In the manuscript, we modified the paragraph in the revised manuscript as below:

As for $\text{Ti}(\text{CO})_8$ (Fig. 2a), the HOMO-LUMO gap of $\text{Ti}(\text{CO})_8$ is 5.47 eV under the same theoretical level of M062x/def2tzvpp. As for $[\text{Ca}(\text{CO})_8]^{2+}$, $[\text{Sc}(\text{CO})_8]$, $[\text{V}(\text{CO})_8]^+$ and $[\text{Cr}(\text{CO})_8]^{2+}$, the HOMO-LUMO gaps are 3.12 eV, 4.28 eV, 6.94 eV and 9.12 eV, respectively. There are no virtual frequencies in all the five $M^q(\text{CO})_8$ ($M^q = \text{Ca}^{2+}, \text{Sc}, \text{Ti}, \text{V}^+$ or Cr^{2+}) complex or ions. The vibrational frequencies of five $M^q(\text{CO})_8$ are listed in the Supporting information (SI-Table S1). Because we do not find the stable conformation of $[\text{Ca}(\text{CO})_7]^{2+}$, we only calculate the corresponding dissociation energies of $M^q(\text{CO})_8 \rightarrow M^q(\text{CO})_7 + \text{CO}$ for $[\text{Sc}(\text{CO})_8]$, $[\text{Ti}(\text{CO})_8]$, $[\text{V}(\text{CO})_8]^+$ and $[\text{Cr}(\text{CO})_8]^{2+}$ (The results are listed in SI-Table S2). All the results are indicating that $M^q(\text{CO})_8$ complex/ions are the minimums on the PESs.

Table S1 The calculated vibrational frequencies (cm^{-1}) of $M^q(\text{CO})_8$ and $M^q(\text{CO})_7$ ($M^q = \text{Ca}^{2+}, \text{Sc}, \text{Ti}, \text{V}^+$ or Cr^{2+}) at M062x/def2tzvpp level

Compounds/ ions	vibrational frequencies
$[\text{Ca}(\text{CO})_8]^{2+}$	43.30, 43.30, 44.66, 44.66, 44.96, 44.96, 51.73, 51.73, 51.73, 58.69, 58.69, 58.69, 197.30, 197.30, 197.30, 207.18, 211.82, 279.90, 279.90, 279.90, 296.30, 296.30, 296.30, 320.23, 320.23, 347.30, 347.30, 347.30, 376.79, 376.79, 376.79, 386.56, 386.56, 386.56, 396.47, 396.47, 1939.24, 1939.24, 1939.24, 1943.70, 1943.70, 1943.70, 1952.20, 2078.08
$[\text{Sc}(\text{CO})_8]$	45.03, 45.03, 67.31, 67.31, 67.31, 69.32, 69.32, 72.31, 72.31, 72.31, 73.58, 73.58, 73.58, 226.94, 226.94, 226.94, 244.64, 262.06, 301.11, 301.11, 301.11, 310.73, 310.73, 310.73, 430.24, 430.24, 451.69, 451.69, 451.69, 455.74, 455.74, 455.74, 459.44, 459.44, 468.02, 468.02, 468.02, 2017.66, 2017.66, 2017.66, 2037.30, 2037.30, 2037.30, 2046.58, 2154.28
$[\text{Sc}(\text{CO})_7]^\dagger$	20.42, 20.48, 51.29, 51.87, 51.91, 58.32, 61.88, 62.09, 66.11, 70.59, 70.68, 253.14, 253.20, 270.23, 288.77, 291.56, 291.58, 324.39, 340.86, 341.74, 341.90, 372.53, 372.60, 414.25, 424.09, 424.10, 431.33, 461.89, 461.91, 485.59, 493.41, 493.43, 1997.61, 1997.61, 2004.84, 2025.94, 2032.12, 2032.12, 2143.29
$\text{Ti}(\text{CO})_8$	45.72, 45.72, 78.65, 78.65, 78.65, 78.09, 78.09, 78.09, 84.15, 84.15, 84.17, 84.17, 84.17, 207.31, 207.31, 207.31, 219.90, 234.97, 234.97, 234.97, 269.58, 286.69, 286.69, 286.69, 471.45, 471.45, 471.45, 477.18, 477.18, 500.36, 500.36, 515.10, 515.10, 516.24, 516.24, 516.24, 2123.43, 2123.43, 2144.97, 2144.97, 2144.97, 2149.43, 2231.66
$\text{Ti}(\text{CO})_7$	26.60, 26.76, 69.62, 73.01, 75.41, 75.45, 78.20, 78.21, 85.82, 89.80, 260.47, 260.51, 285.62, 307.61, 310.91, 310.93, 329.35, 351.33, 351.43, 362.05, 379.31, 379.33, 439.67, 470.02, 470.04, 492.04, 526.23, 526.27, 551.04, 561.04, 561.05, 2102.54, 2108.24, 2108.25, 2141.72, 2164.11, 2164.12, 2235.50
$[\text{V}(\text{CO})_8]^+$	43.90, 43.90, 74.26, 74.26, 74.26, 77.23, 77.23, 77.23, 83.12, 83.12, 83.12, 85.26, 85.26, 115.01, 115.01, 115.01, 135.21, 145.83, 145.83, 145.83, 219.70, 226.80, 226.80, 226.80, 396.68, 396.68, 437.21, 437.21, 437.21, 477.66, 477.66, 477.66, 497.94, 497.94, 505.03, 505.03, 505.03, 2252.88, 2252.88, 2252.88, 2262.96, 2263.33, 2263.33, 2263.33, 2307.85
$[\text{V}(\text{CO})_7]^\dagger$	17.48, 17.95, 75.61, 76.76, 80.08, 80.20, 89.57, 89.76, 99.63, 100.59, 100.65, 235.56, 235.58, 242.35, 278.16, 283.71, 283.78, 293.70, 308.53, 308.62, 325.69, 338.76, 339.00, 422.20, 460.95, 461.02, 493.86, 532.18, 545.68, 562.01, 562.06, 2223.03, 2233.09, 2259.53, 2287.31, 2287.31, 2316.88
$[\text{Cr}(\text{CO})_8]^{2+}$	21.03, 37.11, 37.11, 54.83, 54.83, 54.83, 76.24, 76.24, 76.24, 79.18, 79.18, 83.61, 83.61, 83.61, 109.53, 109.53, 109.53, 125.06, 125.06, 125.06, 181.95, 201.80, 201.81, 201.81, 260.40, 260.40, 369.71, 369.71, 369.71, 397.35, 397.35, 397.35, 437.55, 437.55, 437.55, 443.87, 443.87, 2356.57, 2356.66, 2356.66

$[\text{Cr}(\text{CO})_7]^{2+}$	4.29, 5.74, 72.34, 73.60, 74.49, 74.55, 88.63, 88.71, 97.92, 98.25, 98.29, 185.60, 201.62, 201.67, 207.78, 242.99, 243.06, 244.93, 251.08, 252.19, 263.52, 271.80, 272.08, 371.36, 402.77, 402.95, 428.15, 451.68, 451.69, 490.79, 490.87, 2360.00, 2364.34, 2364.34, 2369.76, 2381.22, 2381.22, 2387.26
--

Table S2 The CO dissociation energies of $M^q(\text{CO})_8$ ($M^q = \text{Sc}^-, \text{Ti}, \text{V}^+ \text{ or } \text{Cr}^{2+}$)^{*} at M062x/def2tzvpp level.

$M(\text{CO})_8 \rightarrow M(\text{CO})_7 + \text{CO}$	$\Delta_r G_m^\ominus$ (kcal/mol)
$[\text{Sc}(\text{CO})_8]^- \rightarrow [\text{Sc}(\text{CO})_7]^- + \text{CO}$	-2.65
$\text{Ti}(\text{CO})_8 \rightarrow \text{Ti}(\text{CO})_7 + \text{CO}$	-15.91
$[\text{V}(\text{CO})_8]^+ \rightarrow [\text{V}(\text{CO})_7]^+ + \text{CO}$	-21.80
$[\text{Cr}(\text{CO})_8]^{2+} \rightarrow [\text{Cr}(\text{CO})_7]^{2+} + \text{CO}$	-17.76

^{*} we do not find the stable conformation of $[\text{Ca}(\text{CO})_7]^{2-}$

3. For completeness, the authors should include $\text{Sc}(\text{CO})_8^-$ and $\text{V}(\text{CO})_8^+$ as well.

We have added the descriptions of $\text{Sc}(\text{CO})_8^-$ and $\text{V}(\text{CO})_8^+$ in the manuscript. Figure 2 has been revised, where all the numbers have been moved into Table 1. The details are listed below. The modifications of Figure 2 are showed in the answers of question 5 below.

1. Therefore, we design a series of O_h -symmetric and singlet-state $M^q(\text{CO})_8$ ($M^q = \text{Ca}^{2+}, \text{Sc}^-, \text{Ti}, \text{V}^+, \text{Cr}^{2+} \text{ or } \text{Ba}^{2+}$) complex/ions theoretically to learn their bonding structures.

2. To reveal the mystery of the state of the ligand-only orbital (a_{2u} orbital), we propose to explain the bonding structures of $M^q(\text{CO})_8$ ($M^q = \text{Ca}^{2+}, \text{Sc}^-, \text{Ti}, \text{V}^+ \text{ or } \text{Cr}^{2+}$) as the singlet '20e-superatom' models based on the comparison in Fig. 1.

3. As for $\text{Ti}(\text{CO})_8$ (Fig. 2a), the HOMO-LUMO gap of $\text{Ti}(\text{CO})_8$ is 5.47 eV under the same theoretical level of M062x/def2tzvpp. As for $[\text{Ca}(\text{CO})_8]^{2-}$, $[\text{Sc}(\text{CO})_8]^-$, $[\text{V}(\text{CO})_8]^+$ and $[\text{Cr}(\text{CO})_8]^{2+}$, the HOMO-LUMO gaps are 3.12 eV, 4.28 eV, 6.94 eV and 9.12 eV, respectively. There are no virtual frequencies in all the five $M^q(\text{CO})_8$ ($M^q = \text{Ca}^{2+}, \text{Sc}^-, \text{Ti}, \text{V}^+ \text{ or } \text{Cr}^{2+}$) complex or ions. The vibrational frequencies of five $M^q(\text{CO})_8$ are listed in the Supporting information (SI-Table S1). Because we do not find the stable conformation of $[\text{Ca}(\text{CO})_7]^{2-}$, we only calculate the corresponding dissociation energies of $M^q(\text{CO})_8 \rightarrow M^q(\text{CO})_7 + \text{CO}$ for $[\text{Sc}(\text{CO})_8]^-$, $[\text{Ti}(\text{CO})_8]$, $[\text{V}(\text{CO})_8]^+$ and $[\text{Cr}(\text{CO})_8]^{2+}$ (The results are listed in SI-Table S2).

4. All the |ONs| are closed to the ideal value of 2.0 |e| in Table 1. We deal with the other four $M^q(\text{CO})_8$ ions ($M^q = \text{Ca}^{2+}, \text{Sc}^-, \text{V}^+ \text{ or } \text{Cr}^{2+}$) in the exactly same pathway to obtain the same configuration of $|1S^2|1P^6|1D^{10}|1F^2|$ based on the jellium model and similar results from AdNDP method.

5. In order to make clear whether the a_{2u} orbital in any $M^q(\text{CO})_8$ ($M^q = \text{Ca}^{2+}, \text{Sc}^-, \text{Ti}, \text{V}^+ \text{ or } \text{Cr}^{2+}$) participates in the formation of eight coordinative bonds, we apply AdNDP method to compare the |ONs| of eight Ti-C orbitals of $\text{Ti}(\text{CO})_8$ with and without the participation of a_{2u} orbital (showed in Fig. 2b and 2c).

6. The similar comparison for the other three $M^q(\text{CO})_8$ ions ($M^q = \text{Sc}^-, \text{V}^+ \text{ or } \text{Cr}^{2+}$) are listed in the supporting information (SI-table S3), where we can obtain the similar conclusions.

7. After dealing with a_{2u} orbitals of the other four $M^q(\text{CO})_8$ ions ($M^q = \text{Ca}^{2+}, \text{Sc}^-, \text{V}^+ \text{ or } \text{Cr}^{2+}$) (Fig. 2d) with the same methods, we also obtain the similar results. The diminutions of |ONs| from 1.96 (1.94/1.95/1.96) to 1.72 (1.71/1.72/1.73) of $[\text{Ca}(\text{CO})_8]^{2-}$ ($[\text{Sc}(\text{CO})_8]^-/[\text{V}(\text{CO})_8]^+/[\text{Cr}(\text{CO})_8]^{2+}$) suggest the both the a_{2u} orbitals contribute around 1/8 components in the coordinative orbital (Table 2).

8. The similar comparison for the other three $M^q(\text{CO})_8$ ions ($M^q = \text{Sc}^-, \text{V}^+ \text{ or } \text{Cr}^{2+}$) are listed in the supporting information (SI-table S3), where we can obtain the similar conclusions.

Table 2. The contribution of the ligand-only a_{2u} orbital in the coordinative bond

	$[\text{Ca}(\text{CO})_8]^{2-}$	$[\text{Sc}(\text{CO})_8]^-$	$\text{Ti}(\text{CO})_8$	$[\text{V}(\text{CO})_8]^+$	$[\text{Cr}(\text{CO})_8]^{2+}$
ONs 1	1.96	1.94	1.94	1.95	1.96
ONs 2	1.72	1.71	1.71	1.72	1.73
Ratio	12.2%	11.9%	11.8%	11.8%	11.7%

|ONs|1 is the occupation number including the contribution of a_{2u} orbital. |ONs|2 is the occupation number without the contribution of a_{2u} orbital.

Ratio is the components of the a_{2u} orbital, which is calculated as:

$$(|\text{ONS}|1 - |\text{ONS}|2) / |\text{ONS}|1$$

Table S3 The comparison of Wiberg Bond Index (WBI) of metal-carbon bond analysis in $\text{M}^q(\text{CO})_8$ and $\text{M}^q(\text{CO})_7$ ($\text{M}^q = \text{Ca}^{2+}$, Sc^+ , Ti , V^+ or Cr^{2+})

M^q	Ca^{2+**}	Sc^+	Ti	V^+	Cr^{2+}
$\text{M}^q(\text{CO})_8$	0.13	0.86	0.88	0.77	0.66
$\text{M}^q(\text{CO})_7$	-	0.97	1.03	0.93	0.81
Ratio	-	0.886/1	0.854/1	0.828/1	0.814/1

*The ratio of the σ -bonding orbital in all the eight orbitals of $\text{Ti}(\text{CO})_8$

** We do not find the stable conformation of $[\text{Ca}(\text{CO})_7]^{2-}$

4. Regarding the introduction, several “metals” should be “transition metals”, as the 18e rule is generally applicable only for transition metals.

We have modified all the relevant descriptions in the manuscript. The details are listed and highlighted as below:

1. The 18-electron rule is a very classic tool for us to understand the structures of a large amount of transition metal complexes.¹ Especially for the homoleptic carbonyl metal complexes, it is very successful to apply Dewar-Chatt-Duncanson (DCD) model² and 18-electron rule to explain the interactions between carbon monoxide and transition metals, such as the seven-coordinate carbonyl cations $[\text{TM}(\text{CO})_7]^+$ ($\text{TM} = \text{V}^+$, Nb^+ and Ta^+) and the eight-coordinated carbonyl anions $[\text{TM}(\text{CO})_8]^-$ ($\text{TM} = \text{Sc}^+$, Y and La^+).
2. There is an interesting inconsistency in understanding the electronic structures of octacarbonyl complexes $\text{M}(\text{CO})_8$ ($\text{M} = \text{Ca}$, Sr and Ba) or cations $[\text{TM}(\text{CO})_8]^+$ ($\text{TM} = \text{Sc}^+$, Y and La^+) on the basis of different chemical bonding theories.
3. As for $\text{TM}(\text{CO})_8$, it has been hypothesized that the eight σ -type orbitals are contributed by both the ligands and the transition metal, where the transition metal possibly adopts a higher-level hybridization (such as the f -type polarization³) for bonding with the eighth carbonyl groups.
4. Therefore, 20-e model (Jellium model) is the most appropriate explanation in understanding the valence bonding structures of $\text{M}(\text{CO})_8$ or $\text{TM}(\text{CO})_8$ complexes and ions.

5. Figure 2 is very crowded and difficult to follow. The authors should clean the figure and put the numbers into a table.

We have modified Figure 2 and removed all the details into Table 1 in the revised manuscript. Here are the detailed modifications:

Fig. 2 (a) AdNDP analysis of $M^q(\text{CO})_8$ ($M^q = \text{Ca}^{2+}, \text{Sc}^+, \text{Ti}, \text{V}^+ \text{ or } \text{Cr}^{2+}$) complex/ions, (b) The detailed AdNDP analysis of (|ON|s) of the five $M^q(\text{CO})_8$ complex/ions with the participation of a_{2u} orbital and (c) without the participation of a_{2u} orbital. (d) The occupied numbers (|ON|s) of $M^q(\text{CO})_8$ complex/ions with/without the participation of a_{2u} orbital.

Table 1 The results of AdNDP analysis and occupied numbers of $M^q(\text{CO})_8$ ($M^q = \text{Ca}^{2+}, \text{Sc}^+, \text{Ti}, \text{V}^+ \text{ or } \text{Cr}^{2+}$)

AdNDP analysis	Orbital type	Occupied numbers (ON s) of orbitals				
		$[\text{Ca}(\text{CO})_8]^{2-}$	$[\text{Sc}(\text{CO})_8]^-$	$\text{Ti}(\text{CO})_8$	$[\text{V}(\text{CO})_8]^+$	$[\text{Cr}(\text{CO})_8]^{2-}$
Localize d MOs of CO	$8 \times \text{LP}(\text{O})$	1.98 e	1.98 e	1.97 e	2.00 e	1.98 e
	$16 \times \pi(\text{CO})$	2.00 e (C: 0.47;O: 1.53)	1.99 e (C: 0.49;O: 1.50)	2.00 e (C: 0.52;O: 1.48)	2.00 e (C: 0.59;O: 1.41)	2.00 e (C: 0.47;O: 1.53)
	$8 \times \sigma(\text{CO})$	2.00 e (C: 0.56;O: 1.44)	2.00 e (C: 0.57;O: 1.43)	2.00 e (C: 0.58;O: 1.42)	2.00 e (C: 0.54;O: 1.46)	2.00 e (C: 0.58;O: 1.42)
Delocalized MOs among ligands and Metal	$1 \times 9\text{c-}2\text{e bonds}$	2.00 e (C: 1.61,Ca: 0.39)	2.00 e (C: 1.54,Sc: 0.43)	2.00 e (C: 1.55,Ti: 0.45)	2.00 e (C: 1.54,V: 0.46)	1.99 e (C: 1.46Cr: 0.53)
	$3 \times 9\text{c-}2\text{e bonds}$	1.99 e (C: 1.63,Ca: 0.36)	1.98 e (C: 1.20,Sc: 0.78)	1.97 e (C: 1.31,Ti: 0.46)	1.98 e (C: 1.08,V: 0.90)	1.99 e (C: 1.60,Cr: 0.39)
	$3 \times 9\text{c-}2\text{e bonds}$	1.99 e (C: 1.51,Ca: 0.48)	1.98 e (C: 1.56,Sc: 0.42)	1.98 e (C: 1.07,Ti: 0.91)	1.98 e (C: 1.53,V: 0.45)	1.98 e (C: 1.75,Cr: 0.23)
	$1 \times 9\text{c-}2\text{e bonds}$	1.98 e (C: 1.98,Ca: 0)	1.98 e (C: 1.98,Sc: 0)	1.97 e (C: 1.97,Ti: 0)	1.98 e (C: 1.98,V: 0)	1.99 e (C: 1.99,Cr: 0)
	$2 \times 17\text{c-}2\text{e bonds}$	2.00 e (C: 1.12,O: 0.35, Ca: 0.53)	1.98 e (C: 0.85,O: 0.29, Sc: 0.86)	2.00 e (C: 0.58;O: 0.22, Ti: 1.20)	2.00 e (C: 0.32,O: 0.12, V: 1.56)	2.00 e (C: 0.17,O: 0.06, Cr: 1.77)

Reviewer #2 (Remarks to the Author):

The authors have analyzed the bonding of the recently introduced octacarbonyl metal complexes $M(\text{CO})_8$. Based on the Jellium model, the stability of these complexes is attributed to satisfying the Jellium model with 20 electrons. In this way, each coordinative bond contains 7/8 of bonding and 1/8 of non-bonding contributions. This non-bonding contribution comes from the a_{2u} orbital, which is extensively discussed in this work. Overall, this manuscript explains the presence of 8 coordinative bonds with a bond order of 7, supporting both valence bond and hybrid bond theories. Its interest and impact to the community, makes me recommend its publication in Communications Chemistry. However, before its acceptance, several points should be either addressed or clarified:

1) $M(\text{CO})_8$ with $M = \text{Ca}, \text{Sr},$ or Ba were initially reported to be neutral species, thus with 18 electrons. However, in this manuscript, the authors first refer to the neutral species, however when referring to the Jellium model, the dianion species are taken into analysis, thus with 20 electrons. The authors should clearly specify the reason why they go from the initial neutral species, with O_h symmetry and a triplet electronic ground state, to the dianions, with O_h symmetry as well, but closed shell singlet state.

This manuscript is trying to explain the phenomenon based on a series of experiments of Prof Zhou's. Their previous reports (Science 361, 912-916 (2018)) are about the alkaline earth complexes $M(\text{CO})_8$ ($M = \text{Ca}, \text{Sr},$ or Ba), which are the neutral species adopting O_h symmetry but with triplet states. The two electrons in the e_g orbitals spin remain parallel. It is traditional to apply Dewar-Chat-Duncanson (DCD) model and 18-electron rule to explain the interactions between carbon monoxide and transition metals. So in our introduction, we first introduce $M(\text{CO})_8$ as neutral species. But it is curious for us that what's the role of the ligand-only a_{2u} orbital in the 10 valence orbitals?

In our study, we emphasize the 20-e jellium model is an appropriate tool to understand the valence orbital of $M^q(\text{CO})_8$ ($M^q = \text{Ca}^{2-}, \text{Sc}^-, \text{Ti}, \text{V}^+, \text{Cr}^{2+}$ or Ba^{2-}) complex/ions, where 20 is the magic number of Jellium model. Furthermore, all the 10 valence orbitals ($a_{1g} + 3t_{1u} + 3t_{2g} + a_{2u} + 2e_g$) can be fulfilled with maximum 20 electrons as the singlet states for the octacarbonyl metal complexes or ions.

Therefore, we theoretically design all the O_h -symmetric octacarbonyl metal complexes or ions with 20-e on purpose. Actually, there are no differences of the valence orbitals among the experimental alkaline earth complexes $M(\text{CO})_8$ ($M = \text{Ca}, \text{Sr},$ or Ba) with 18-e and the theoretical designed O_h -symmetric $[\text{BaBe}_8]^{2-}$ and $M^q(\text{CO})_8$ ($M^q = \text{Ca}^{2-}, \text{Sc}^-, \text{Ti}, \text{V}^+, \text{Cr}^{2+}$ or Ba^{2-}). All the valence orbitals of such complexes or ions are similar with each other, which are $a_{1g} + t_{1u} + t_{2g} + a_{2u} + e_g$ orbitals. These orbitals can be fulfilled with 20 electrons. The designed analogical comparison of $[\text{BaBe}_8]^{2-}$ and $[\text{Ba}(\text{CO})_8]^{2-}$ is in order to demonstrate Jellium model can be applied to analyse the bonding structure of $[\text{Ba}(\text{CO})_8]^{2-}$.

In the revised manuscript, we add the explanation for the 'different' octacarbonyl metal complexes or ions. Here are details:

On the basis of the experimental studies, all the 10 valence orbitals ($a_{1g} + 3t_{1u} + 3t_{2g} + a_{2u} + 2e_g$) can be fulfilled with maximum 20 electrons as the singlet states for the O_h -symmetric octacarbonyl metal complexes or ions. Therefore, it is attractive for us to understand the state of the ligand-only orbital (a_{2u} orbital) essentially.

For another aspect, the octacarbonyl metal complex or ion is with O_h symmetries, where the octa-coordination field contributed by the positive center metal and eight ligands can be approximately viewed as a homogeneous spherical field or a Jellium model^{12,13}. Therefore, we design a series of O_h -symmetric and singlet-state $M^q(\text{CO})_8$ ($M^q = \text{Ca}^{2-}, \text{Sc}^-, \text{Ti}, \text{V}^+, \text{Cr}^{2+}$ or Ba^{2-}) complex/ions theoretically to learn their bonding structures. Such $M^q(\text{CO})_8$ complex/ions are 20-e closed-shell molecules based on the configuration³, which is exactly satisfying the magic stability of Jellium model^{12,13}.

2) Regarding the Jellium model, why is 2S not occupied? Is it higher in energy than 1F2?

First, the energy level of 2S and 1F orbitals are close to each other. But 1F orbitals are more susceptible to the ligands. Secondly, it is a cubic field in all the $M^q(\text{CO})_8$ complexes or ions, which affect the F orbital in the coordination. The 7-degenerate F orbitals can be simplified as Figure 1 (a), where the a_{2u} -symmetric F_{xyz} orbital matches the orbital symmetry in the cubic field of coordination. Therefore,

as a bonding orbital, the energy of $1F_{xyz}$ orbital is lower than $2S$ orbital caused by the splitting of F orbitals affected by the cubic field. As showed in Figure1 (b) below, the a_{2u} orbital of $[\text{Ba}(\text{CO})_8]^{2-}$ is showed below, where the eight ligands coordinate with Ba^{2-} matching of orbitals symmetry.

In order to clarify our viewpoints, we also optimized $[\text{U}(\text{CO})_8]^{4+}$ to understand the splitting of the f orbitals in Figure 2, where the 5f atomic orbitals of U^{4+} are contributed in the bonding structure. Obviously, all the MOs are similar with that of $[\text{Ba}(\text{CO})_8]^{2-}$, except for the orbital energies. The energy of a_{2u} orbital is lower than d -type orbital because of the splitting of 5f AOs of U^{4+} .

So in the manuscript, we describe the energy level as Figure 1 showed, where the electron first occupied a_{2u} -symmetric $1F_{xyz}$ orbital rather than a_{1g} -symmetric $2S$ orbital.

On the basis of the suggestion, we add some explanations in the manuscript as:

It should be noticed that the a_{2u} orbital is an $1F$ orbital rather than $2S$ orbital, where the energy of $1F$ orbital is in the middle of the two groups $1D$ orbitals.

Based on the calculation, a_{2u} orbital adopts the f symmetry as showed in Fig. 1. Furthermore, there is a cubic field in all the $\text{M}^q(\text{CO})_8$ complexes or ions, which affect the 7-degenerate $1F$ orbitals in the coordination, in which the a_{2u} -symmetric $1F_{xyz}$ orbital matches the orbital symmetry in the cubic field of coordination. Therefore, the energy of $1F_{xyz}$ orbital is lower than $2S$ orbital caused by the splits of F orbitals (Figure S1 in the supporting information). The sequences of energy levels of the orbitals are the results of the splitting of $1D$ and $1F$ orbitals.

Figure 1 The schematic diagram of 7-degenerate F orbitals (a) and the a_{2u} orbital of $[\text{Ba}(\text{CO})_8]^{2-}$ (b).

Figure 2 The MOs of $[\text{U}(\text{CO})_8]^{4+}$.

3) Very recently, Poater and Solà have extended the Jellium model to open-shell half-filled systems (Chem. Comm. 2019, 55, 5559-5562). Thus, and related to point 1 above, could this new rule be applied to neutral $M(\text{CO})_8$ systems with triplet ground state? The two highest e_g ($1D(2)$) orbitals are half-filled, thus fulfilling this new rule. This would also justify the stability of these systems, not being necessary to go into the dianions, and to further support the theory presented in this manuscript. Both sets of systems could be enclosed and discussed, both of them supported by Jellium model, the one with 20 electrons with the closed-shell model, whereas that with 18 electrons with the open-shell model.

We have read this paper recently and we don't think the neutral $M(\text{CO})_8$ satisfies the open-shell Jellium model. In Sola's reports, they have hypothesized the rule of ' $2N^2 + 2N + 1$ with $S = N + 1/2$ ', where the magic numbers of valence electrons have been defined as 1 ($S=1/2$), 5 ($S=3/2$), 13 ($S=5/2$), 19 ($S=7/2$).... All the results are calculated from the highest energy level of valence electrons half-filled the same-spin electrons, which is defined as the open-shell model by the authors. We should notice that there is not any orbital splitting in their examples, where all the degenerate D or F orbitals are degeneracy. All the inner orbitals are fulfilled with electrons. At the meantime, all the outer orbitals are half-filled. We consider this is the previous condition for the ' $2N^2 + 2N + 1$ with $S = N + 1/2$ ' rule.

But in neutral $M(\text{CO})_8$, 5-degenerate 1D orbitals split to two groups. The highest $1D_{x^2-y^2}$ and $1D_{z^2}$ orbitals are corresponding to the two $d \rightarrow \pi^*$ backdonation bond, which still belong to the inner 1D orbitals. The number of electrons of triplet $M(\text{CO})_8$ are 18, which is also not satisfying the magic number of open-shell Jellium model. So we design our systems as a 20-e closed-shell Jellium model, where all the theoretical models as the singlet state have the common electronic configurations.

For another aspect, there are 10 super orbitals in total based on whatever closed-shell or open-shell Jellium model. 20 is the maximum electron number the 10 super orbitals can accommodate, which is different from the traditional 18-e rule or Dewar-Chat-Duncanson (DCD) model, where 9 AOs of center metal can only accommodate 18 electrons. In addition, there are no essentially differences for the role of the a_{2u} orbital, which is still a ligand-only orbital but contribute 1/8 components in each coordination bond. As for the closed-shell or open-shell Jellium model, we think open-shell model is like the Hund rule applied in Jellium model, which is a new and interesting viewpoint to define the magic stability of a kind of special cluster.

We compare the open-shell model pointed by Sola with the neutral $M(\text{CO})_8$ in this Figure below. Figure (a) is snapshot from Sola's paper (Chem. Comm. 2019, 55, 5559-5562). Figure (b) is the electronic configuration of neutral $M(\text{CO})_8$, which can be simplified as $|1S^2|1P^6|1D^8|1F^2|$.

Figure for question 3: the comparison of the the open-shell model pointed by Sola with the neutral $M(\text{CO})_8$

4) BaBe8 is first presented as neutral (title), but the whole discussion is based on the dianion. Please correct accordingly.

Yes. We have modified the relevant contents in the revised paper based on the suggestions.

The abstract: By comparing $[\text{Ba}(\text{CO})_8]^{2-}$ and metal clusters of $[\text{BaBe}_8]^{2-}$ analogically, we demonstrate Jellium model is also a useful tool in understanding the electronic structures of $M^q(\text{CO})_8$ ($M^q = \text{Ca}^{2-}, \text{Sc}^-, \text{Ti}, \text{V}^+, \text{Cr}^{2+}$ or Ba^{2-}) rather than only for metal clusters.

The subtitle: The similarity of $[\text{Ba}(\text{CO})_8]^{2-}$ and $[\text{BaBe}_8]^{2-}$ (cluster)

The title of Fig 1: The molecular orbitals listed on the basis of Jellium model of singlet $[\text{Ba}(\text{Be})_8]^{2-}$ (left) and singlet $[\text{Ba}(\text{CO})_8]^{2-}$ (right).

5) If all calculations have been performed with the M06-2X functional, why for $\text{Ti}(\text{CO})_8$ BP86 is used? (second paragraph in page 3) Maybe this is an error, as in the Supporting Information only M06-2X has been used.

Yes. It is an error in the previous manuscript. Now we have modified in the revised paper:

As for $\text{Ti}(\text{CO})_8$ (Fig. 2a), the HOMO-LUMO gap of $\text{Ti}(\text{CO})_8$ is 5.47 eV under the same theoretical level of M062x/def2tzvpp.

6) In the caption of Figure 2, please add also the colors used for each of the three systems: red, black and blue.

Figure 2 is very crowded and difficult to follow. Now we have modified Figure 2 and listed all the details into Table 1. Here are the detailed modifications:

Fig. 2 (a) AdNDP analysis of $M^q(\text{CO})_8$ ($M^q = \text{Ca}^{2-}, \text{Sc}^-, \text{Ti}, \text{V}^+, \text{or Cr}^{2+}$) complex/ions, (b) The detailed AdNDP analysis of (|ON|s) of the five $M^q(\text{CO})_8$ complex/ions with the participation of a_{2u} orbital and (c) without the participation of a_{2u} orbital. (d) The occupied numbers (|ON|s) of $M^q(\text{CO})_8$ complex/ions with/without the participation of a_{2u} orbital.

Table 1 The results of AdNDP analysis and occupied numbers of $M^q(\text{CO})_8$ ($M^q = \text{Ca}^{2-}, \text{Sc}^-, \text{Ti}, \text{V}^+, \text{or Cr}^{2+}$)

AdNDP analysis	Orbital type	Occupied numbers (ON s) of orbitals				
		$[\text{Ca}(\text{CO})_8]^{2-}$	$[\text{Sc}(\text{CO})_8]^-$	$\text{Ti}(\text{CO})_8$	$[\text{V}(\text{CO})_8]^+$	$[\text{Cr}(\text{CO})_8]^{2-}$
Localize d MOs of CO	8×LP(O)	1.98 e	1.98 e	1.97 e	2.00 e	1.98 e
	16×π(CO)	2.00 e (C: 0.47;O: 1.53)	1.99 e (C: 0.49;O: 1.50)	2.00 e (C: 0.52;O: 1.48)	2.00 e (C: 0.59;O: 1.41)	2.00 e (C: 0.47;O: 1.53)
	8×σ(CO)	2.00 e (C: 0.56;O: 1.44)	2.00 e (C: 0.57;O: 1.43)	2.00 e (C: 0.58;O: 1.42)	2.00 e (C: 0.54;O: 1.46)	2.00 e (C: 0.58;O: 1.42)
Delocaliz ed MOs among ligands and	1×9c-2e bonds	2.00 e (C: 1.61,Ca: 0.39)	2.00 e (C: 1.54,Sc: 0.43)	2.00 e (C: 1.55,Ti: 0.45)	2.00 e (C: 1.54,V: 0.46)	1.99 e (C: 1.46Cr: 0.53)
	3×9c-2e bonds	1.99 e (C: 1.63,Ca: 0.36)	1.98 e (C: 1.20,Sc: 0.78)	1.97 e (C: 1.31,Ti: 0.46)	1.98 e (C: 1.08,V: 0.90)	1.99 e (C: 1.60,Cr: 0.39)
	3×9c-2e bonds	1.99 e	1.98 e	1.98 e	1.98 e	1.98 e

Metal		(C: 1.51,Ca: 0.48)	(C: 1.56,Sc: 0.42)	(C: 1.07,Ti: 0.91)	(C: 1.53,V: 0.45)	(C: 1.75,Cr: 0.23)
1×9c-2e bonds	with t_{2g} symmetry					
	F bond with a_{2u} symmetry	1.98 e	1.98 e	1.97 e	1.98 e	1.99 e
2×17c-2e bonds	$Dz^2, D_{x^2-y^2}$ bonds	2.00 e	1.98 e	2.00 e	2.00 e	2.00 e
	with e_g symmetry	(C: 1.12,O: 0.35, Ca: 0.53)	(C: 0.85,O: 0.29, Sc: 0.86)	(C: 0.58;O: 0.22; Ti: 1.20)	(C: 0.32,O: 0.12; V: 1.56)	(C: 0.17,O: 0.06, Cr: 1.77)

7) Can the authors justify the reason why 1D(4) orbitals can be either lower (Figure 3) or higher in energy than 1F(2) (Figure 1)? How does the change of M affect these trends?

Fig. 3 (a) The MO diagram (σ orbitals) of $M^n(CO)_8$ ($M^n=Ca^{2+}, Sc^+, Ti, V^+, Cr^{2+}$ or Ba^{2+}); (b) The $d-\pi^*$ backdonation bonds of $M(CO)_8$ complex; (c) The configuration as jellium model by viewing $M^n(CO)_8$ as a '20-e superatom'.

This is an error in the figure 3 previously. We made an error to distinguish the energy level between $1D_{x^2-y^2}/1D_{z^2}$ orbitals and $1F^2$ orbital. This figure is a schematic figure, where we hope to show the splitting of 5-degenerate 1D orbitals. The 3-degenerate orbitals are σ -type orbitals in the coordination. The other 2-degenerate $1D_{x^2-y^2}$ and $1D_{z^2}$ orbitals are corresponding to the $d \rightarrow \pi^*$ backdonation bonds. Now we modified Figure 3 in the revised manuscript. The results are consistent with Zhou's reports (Science 361, 912-916 (2018)). In the revised paper, we summarize all the energy levels for the five compounds in the Table and put them into the Table S4 of supporting information. The results indicate the energy of 1F orbital is between 3-degenerate 1D orbitals ($1D_{xy, xz, yz}$) and 2-degenerate 1D orbitals ($1D_{x^2-y^2}$ and $1D_{z^2}$).

Table S4 The energies (eV) of super orbitals of $M^n(CO)_8$ ($M^n=Ca^{2+}, Sc^+, Ti, V^+$ or Cr^{2+}) based on the Jellium model.

orbitals	$[Ca(CO)_8]^{2+}$	$[Sc(CO)_8]^+$	$Ti(CO)_8$	$[V(CO)_8]^+$	$[Cr(CO)_8]^{2+}$
1S	-6.63	-11.53	-16.24	-20.60	-24.67
1P	-5.14	-9.52	-14.00	-18.42	-22.74
1D ($1D_{xy, xz, yz}$)	-4.43	-8.65	-13.09	-17.44	-21.80
1F ($1F_{xyz}$)	-2.91	-6.20	-10.04	-14.55	-19.43
1D ($1D_{x^2-y^2, z^2}$)	3.48	-1.44	-6.71	-12.16	-9.71

In our opinion, the energy level of $|1F|^2$ is lower than that of $|1D|^4$, which is caused by the splitting of D orbitals and F orbitals in the cubic field. In the Jellium model, the energies of 1D orbitals are lower than that of 1F orbitals. But the 5-degenerate 1D orbitals split to two groups in the coordination, including 3-degenerate σ -type bonding orbitals with lower energies and 2-degenerate anti-bonding orbitals with higher energies. As for 1F orbital, we have analyzed in Fig. S1, where the splitting of 1F orbitals results in the $1F_{xyz}$ orbital as the non-bonding orbital, which takes part in the coordination as the ligand-only orbital.

And we also add the explanation for the sequences of the orbitals in the revised manuscript:

Therefore, the 15 MOs should be composed as 7 bonding, 1 non-bonding, and 7 anti-bonding orbitals filled by 16 electrons successively. In the 15 MOs, the 7 bonding orbitals are corresponding to the a_{1g} , t_{1u} and t_{2g} orbitals. The non-bonding orbital is exactly corresponding to the ligand-only a_{2u} orbital. This is also consistent with the sequence of energies of orbital levels, where the energy of $1F_{xyz}$ orbital as a non-bonding orbital is between 3-degenerate 1D orbitals ($1D_{xy, xz, yz}$) and 2-degenerate 1D orbitals ($1D_{x^2-y^2}$ and $1D_{z^2}$).

8) Finally, several misprints along the manuscript should be corrected.

We have rewritten most part of the manuscript based on all the comments. Here are the details, where all the modifications are highlighted in the table

Page 1	
The recently reported octacarbonyl metal complexes $M(CO)_8$ appear very interesting bonding structures. In these compounds, there are 8 coordinative bonds (CO→Metal) and 2 backdonation bonds ($d \rightarrow \pi^*$) based on the valence bond (VB) theory. However, there are only seven valence orbitals (d^3sp^3 hybridization with the bond order of 7) participating in forming σ -type bonds. This is an interesting but a common phenomenon that the bond order is 7 but with accommodating 8 LPs of ligands in forming octa-coordinated complex. Here we demonstrate Jellium model is also a useful tool in understanding the electronic structure of $M(CO)_8$ not only for metal clusters. By applying Jellium model, the magic stability of $M(CO)_8$ complex is because the structure is satisfying the '20-electron' rule (20-e is the magic number of the Jellium model) by comparing $M(CO)_8$ and metal clusters of $BaBe_8$ analogically. As for the 8 coordinative bonds, each coordinative bond contains 1/8 components of the non-bonding orbital (or $ 1F^2\rangle$ orbital in Jellium model) in $M(CO)_8$ complexes.	The recently reported octacarbonyl metal complexes $M(CO)_8$ ($M = Ca, Sr$ and Ba) appear very interesting bonding structures. In these compounds, there are 8 coordinative bonds (CO→Metal) and 2 backdonation bonds ($d \rightarrow \pi^*$) based on the valence bond (VB) theory. However, there are only seven valence orbitals (d^3sp^3 hybridization with the bond order of 7) participating in forming σ -type bonds. This is an interesting but a common phenomenon that the bond order is 7 but with accommodating 8 lone pairs (LPs) of ligands in forming octa-coordinated complexes/ions. By comparing $[Ba(CO)_8]^{2-}$ and metal clusters of $[BaBe_8]^{2-}$ analogically, we demonstrate Jellium model is also a useful tool in understanding the electronic structures of $M^q(CO)_8$ ($M^q=Ca^{2+}, Sc^+, Ti, V^+, Cr^{2+}$ or Ba^{2+}) rather than only for metal clusters. By applying Jellium model, the magic stabilities of such $M^q(CO)_8$ complex or ions are because the structure is satisfying the '20-electron' rule (20-e is the magic number of the Jellium model). As for the 8 coordinative bonds, each coordinative bond contains 1/8 components of the non-bonding orbital (or $ 1F^2\rangle$ orbital in Jellium model) in $M^q(CO)_8$ complexes or ions
Page 2	
The 18-electron rule is a very classic tool for us to understand the structures of a large amount of metal complexes. ¹ Especially for the homoleptic carbonyl metal complexes, it is very successful to apply Dewar-Chartt-Duncanson (DCD) model ² and 18-electron rule to explain the interactions between carbon monoxide and metals, such as the seven-coordinate carbonyl complexes $[M(CO)_7]^+$ ($M=V^+, Nb^+$ and Ta^+) and the eight-coordinated carbonyl complexes $[M(CO)_8]^+$ ($M=Sc^+, Y^+$ and La^+). ³⁻⁵ The recent experimentally produced alkaline earth metal complexes $M(CO)_8$ ($M = Ca, Sr$ and Ba) are satisfying the DCD model and 18-electron rule, ^{6,7} including 8 degenerate $\sigma_{CO \rightarrow M}$ coordinative bonds and 2 $(n-1)d \rightarrow \pi^*$ backdonation bonds from alkali metal to the ligands. ⁸	The 18-electron rule is a very classic tool for us to understand the structures of a large amount of transition metal complexes. ¹ Especially for the homoleptic carbonyl metal complexes, it is very successful to apply Dewar-Chartt-Duncanson (DCD) model ² and 18-electron rule to explain the interactions between carbon monoxide and transition metals, such as the seven-coordinate carbonyl cations $[TM(CO)_7]^+$ ($TM=V^+, Nb^+$ and Ta^+) and the eight-coordinated carbonyl anions $[TM(CO)_8]^-$ ($TM=Sc^-, Y^-$ and La^-). ³⁻⁵ The recent experimentally produced alkaline earth metal complexes $M(CO)_8$ ($M = Ca, Sr$ and Ba) are satisfying the DCD model and 18-electron rule, ^{6,7} including 8 degenerate $\sigma_{CO \rightarrow TM}$ coordinative bonds and 2 $(n-1)d \rightarrow \pi^*$ backdonation bonds from alkali earth metal to the ligands. ⁸
There is an interesting inconsistency in understanding the electronic structures of octacarbonyl complexes $M(CO)_8$ ($M = Ca, Sr$ and Ba) or cations $[M(CO)_8]^+$ ($M=Sc^+, Y^+$ and La^+) on the basis of different chemical bonding theories. It should be noted that all the structures adopt O_h symmetry. Therefore, on the basis of valence bond (VB) theory ⁹ , there are eight degenerate coordination bonds and two π -backdonation bonds between metal and ligands in the series $M(CO)_8$ complexes, where eight carbonyl groups as Lewis base provide eight lone pairs	There is an interesting inconsistency in understanding the electronic structures of octacarbonyl complexes $M(CO)_8$ ($M = Ca, Sr$ and Ba) or cations $[TM(CO)_8]^+$ ($TM=Sc^+, Y^+$ and La^+) on the basis of different chemical bonding theories. It should be noted that all the structures adopt O_h symmetry. Therefore, on the basis of valence bond (VB) theory ⁹ , there are eight degenerate coordination bonds and two π -backdonation bonds between metal and ligands in the series $M^q(CO)_8$ complexes or ions, where eight carbonyl groups provide eight lone pairs (LPs) of

(LPs) of electrons to the metal. However, based on the hybrid orbital (HO) theory¹⁰, only seven empty degenerate orbitals can be provided by the metal by adopting d^3sp^3 hybridization. Therefore, the bond orders are inconsistent on the basis of the two classic theories. It is difficult to arrange the 16 LPs in forming an eight-coordinated complex with the bond order of 7. It has been hypothesized that the eight σ-type orbitals are contributed by both the ligands and the metal, where the metal possibly adopts a higher-level hybridization (such as the f-type polarization³) for bonding with the eighth carbonyl groups. It is a reasonable viewpoint to understand the inconsistency of VB and HO theories, where the bond orders are both 8.	electrons to the center metal. However, based on the hybrid orbital (HO) theory¹⁰, only seven empty degenerate orbitals can be provided by the d^2sp^3 hybridized metal. Therefore, the bond orders are inconsistent on the basis of the two classic theories. It is difficult to arrange the 16 LPs in forming an eight-coordinated complex with the bond order of 7. As for $\text{TM}(\text{CO})_8$, it has been hypothesized that the eight σ-type orbitals are contributed by both the ligands and the transition metal, where the transition metal possibly adopts a higher-level hybridization (such as the f-type polarization³) for bonding with the eighth carbonyl groups. It is a reasonable viewpoint to understand the inconsistency of VB and HO theories, where the bond orders are both equal to 8.
Additionally, Zhou's research pointed out that the molecules adopt O_h symmetry with the valence electron configuration of $a_{1g}^2 t_{1u}^6 t_{2g}^6 a_{2u}^2 e_g^2$, where the a_{2u} orbital is explained as a ligand-only orbital, which is satisfying the 18-electron rule perfectly^{3,8}. The electrons distributed in a_{2u} orbital is stabilized by the field effect of the metal on the ligand cage.^{3,11} Besides the ligand-only a_{2u} orbital, the other MOs ($a_{1g} + 3t_{1u} + 3t_{2g} + 2e_g$) including 7 σ-donation bonds ($a_{1g} + 3t_{1u} + 3t_{2g}$) and 2 π-backdonation bonds ($2e_g$) can accommodate 18 valence electrons of $\text{M}(\text{CO})_8$ complex perfectly,⁸ which is also quite reasonable to understand the bonding structure of $\text{M}(\text{CO})_8$ compounds or ions. However, it is attractive for us to understand the state of the ligand-only orbital (a_{2u} orbital) essentially. $\text{M}(\text{CO})_8$ complex is a 20-e closed-shell molecule based on the configuration³, which is exactly satisfying the magic stability in Jellium model^{12,13}. Is it possible the ligand-only orbital as a component participating in the eight coordinative bonds to form a 20-electron configuration for its valence electrons?	Additionally, Zhou's research pointed out that $\text{M}(\text{CO})_8$ complexes or ions adopt O_h symmetry with the valence electron configuration of $a_{1g}^2 t_{1u}^6 t_{2g}^6 a_{2u}^2 e_g^2$, where the a_{2u} orbital is explained as the ligand-only orbital, which is satisfying the 18-electron rule perfectly^{3,8}. The electrons in a_{2u} orbital are stabilized by the field effect of the metal on the ligand cage.^{3,11} Besides the ligand-only a_{2u} orbital, the other 9 MOs ($a_{1g} + 3t_{1u} + 3t_{2g} + 2e_g$) including 7 σ-donation bonds ($a_{1g} + 3t_{1u} + 3t_{2g}$) and 2 π-backdonation bonds ($2e_g$) can accommodate 18 valence electrons of $\text{M}(\text{CO})_8$ complex perfectly,⁸ which is also quite reasonable to understand the bonding structure of $\text{M}(\text{CO})_8$ complexes or ions. On the basis of the experimental studies, all the 10 valence orbitals ($a_{1g} + 3t_{1u} + 3t_{2g} + a_{2u} + 2e_g$) can be fulfilled with maximum 20 electrons as the singlet states for the O_h-symmetric octacarbonyl metal complexes or ions. Therefore, it is attractive for us to understand the state of the ligand-only orbital (a_{2u} orbital) essentially. For another aspect, the octacarbonyl metal complex or ion is with O_h symmetries, where the octa-coordination field contributed by the positive center metal and eight ligands can be approximately viewed as a homogeneous spherical field or a Jellium model^{12,13}. Therefore, we design a series of O_h-symmetric and singlet-state $\text{M}^q(\text{CO})_8$ ($\text{M}^q = \text{Ca}^{2+}, \text{Sc}^+, \text{Ti}, \text{V}^+, \text{Cr}^{2+}$ or Ba^{2+}) complex/ions theoretically to learn their bonding structures. Such $\text{M}^q(\text{CO})_8$ complex/ions are 20-e closed-shell molecules based on the configuration³, which is exactly satisfying the magic stability of Jellium model^{12,13}. We are curious that whether the ligand-only a_{2u} orbital as a component participate in the eight coordinative bonds to form a 20-electron configuration for its valence electrons?
As for the analogical octa-coordinative complex, the O_h-symmetric $[\text{BaBe}_8]^{2-}$ are optimized at M06-2X/def2tzvpp level of theory^{19,20} in Gaussian 09²¹ to obtain the singlet electronic ground state with the valence electron configuration $a_{1g}^2 t_{1u}^6 t_{2g}^6 a_{2u}^2 e_g^4$ with the HOMO-LUMO gap of 2.77 eV. The electrons fulfil the valence orbital following the configuration of $[1S^2 1P^6 1D^6 1F^2 1D^4]$, where D orbitals are split into two groups as ($1D_{xy}, 1D_{yz}$ and $1D_{xz}$) and ($1D_{x^2-y^2}$ and $1D_{z^2}$). It should be noticed that the a_{2u} orbital adopts the f symmetry as showed in Fig. 1. In the configuration, the number of valence electrons just equals to the magic number '20' appearing the magical stability.	As for the analogical octa-coordinative complex, the O_h-symmetric $[\text{BaBe}_8]^{2-}$ are optimized at M06-2X/def2tzvpp level of theory^{19,20} in Gaussian 09²¹ to obtain the singlet electronic ground state. The valence electron configuration of $[\text{BaBe}_8]^{2-}$ is $a_{1g}^2 t_{1u}^6 t_{2g}^6 a_{2u}^2 e_g^4$ with the HOMO-LUMO gap of 2.77 eV. Based on the Jellium model, the electrons fulfil the 10 valence orbitals following the configuration of $[1S^2 1P^6 1D^6 1F^2 1D^4]$, where D orbitals are split into two groups as ($1D_{xy}, 1D_{yz}$ and $1D_{xz}$) and ($1D_{x^2-y^2}$ and $1D_{z^2}$). It should be noticed that the a_{2u} orbital is an 1F orbital rather than 2S orbital, where the energy of 1F orbital is in the middle of the two groups 1D orbitals.

	Based on the calculation, a_{2u} orbital adopts the f symmetry as showed in Fig. 1. Furthermore, there is a cubic field in all the $M^q(\text{CO})_8$ complexes or ions, which affect the 7-degenerate 1F orbitals in the coordination, in which the a_{2u}-symmetric 1F_{xyz} orbital matches the orbital symmetry in the cubic field of coordination. Therefore, the energy of 1F_{xyz} orbital is lower than 2S orbital caused by the splits of F orbitals (Figure S1 in the supporting information). The sequences of energy levels of the orbitals are the results of the splitting of 1D and 1F orbitals. In the configuration, the number of valence electrons just equals to the magic number '20' appearing the magical stability.
Page 3	
The comparison of the molecular orbitals of $[\text{Ba}(\text{CO})_8]^{2-}$ and $[\text{Ba}(\text{Be})_8]^{2-}$ are showed in Fig. 1. Although $[\text{Ba}(\text{CO})_8]^{2-}$ is not a metal cluster, its configuration can be similarly described as jellium model as $[1S^2]1P^6]1D^6]1F^2]1D^4]$. As for $[\text{Ba}(\text{CO})_8]^{2-}$, a_{1g}, t_{1u} and t_{2g} orbitals are corresponding to the 1S, 1P orbitals and $1D_{xy}/1D_{yz}/1D_{xz}$ super orbitals in Jellium model. The e_g MOs are corresponding to the $1D_{x^2-y^2}$ and $1D_{z^2}$ orbitals as two π-type orbitals (or $5d_{x^2-y^2}$ and $5d_{z^2}$ AOs of barium), which are the two $d \rightarrow \pi^*$ bonds donated from 5d AOs of Ba to the 2p AOs of eight CO groups. The ligand-only a_{2u} orbital composed of eight carbonyl groups is 1F super orbital based on the diagram in Fig. 1. The similarity of $[\text{Ba}(\text{CO})_8]^{2-}$ and $[\text{BaBe}_8]^{2-}$ indicate we can understand the complex as a '20e-superatom' based on Jellium model (20 is a magic number suggesting the structural stability). For another aspect, we apply MO theory to understand the eight coordinative bonds in $[\text{Ba}(\text{CO})_8]^{2-}$. There should be 15 MOs combined by 7 valence AOs of Ba atom (d^3sp^3) and 8 AOs of LPs donated from 8 (CO) groups in forming eight $\sigma(\text{Ba-C})$ bonds based on MO theory. Therefore, the 15 MOs should be composed as 7 bonding, 1 non-bonding, and 7 anti-bonding orbitals filled by 16 electrons successively. In the 15 MOs, the 7 bonding orbitals are corresponding to the a_{1g}, t_{1u} and t_{2g} orbitals. The non-bonding orbital is exactly corresponding to the ligand-only a_{2u} orbital.	The comparison of the molecular orbitals of $[\text{Ba}(\text{CO})_8]^{2-}$ and $[\text{Ba}(\text{Be})_8]^{2-}$ are showed in Fig. 1. Although $[\text{Ba}(\text{CO})_8]^{2-}$ is not a metal cluster, its configuration can be similarly described as $[1S^2]1P^6]1D^6]1F^2]1D^4]$ based on the Jellium model. Furthermore, a_{1g}, t_{1u} and t_{2g} orbitals of $[\text{Ba}(\text{CO})_8]^{2-}$ are corresponding to the 1S, 1P orbitals and $1D_{xy}/1D_{yz}/1D_{xz}$ super orbitals. The e_g MOs are corresponding to the $1D_{x^2-y^2}$ and $1D_{z^2}$ orbitals as two π-type orbitals, corresponding to the two $d \rightarrow \pi^*$ bonds donated from 5d AOs of Ba ($5d_{x^2-y^2}$ and $5d_{z^2}$ AOs) to the 2p AOs of eight CO groups. The ligand-only a_{2u} orbital composed of eight carbonyl groups is also defined as the 1F super orbital based on the diagram in Fig. 1. The similarity of $[\text{Ba}(\text{CO})_8]^{2-}$ and $[\text{BaBe}_8]^{2-}$ indicate we can understand the valence orbitals of $[\text{Ba}(\text{CO})_8]^{2-}$ as a '20e-superatom' based on Jellium model. For another aspect, we apply MO theory to understand the eight σ-type coordinative bonds in $[\text{Ba}(\text{CO})_8]^{2-}$. There should be 15 MOs contributed by 7 valence AOs of Ba atom (d^3sp^3) and 8 AOs of LPs donated from 8 (CO) groups in forming eight $\sigma(\text{Ba-C})$ bonds based on MO theory. Therefore, the 15 MOs should be composed as 7 bonding, 1 non-bonding, and 7 anti-bonding orbitals filled by 16 electrons successively. In the 15 MOs, the 7 bonding orbitals are corresponding to the a_{1g}, t_{1u} and t_{2g} orbitals. The non-bonding orbital is exactly corresponding to the ligand-only a_{2u} orbital. This is also consistent with the sequence of energies of orbital levels, where the energy of 1F_{xyz} orbital as a non-bonding orbital is between 3-degenerate 1D orbitals ($1D_{xy}$, xz, yz) and 2-degenerate 1D orbitals ($1D_{x^2-y^2}$ and $1D_{z^2}$).
Page 4	
The 20-electron model for $M(\text{CO})_8$ To reveal the mystery of the state of the ligand-only orbital (a_{2u} orbital), we propose to explain the bonding structure of $[\text{M}(\text{CO})_8]$ complexes as a singlet '20e-superatom' model based on the comparisons in Fig. 1. The typical molecules selected as $\text{Ti}(\text{CO})_8$, $[\text{Ca}(\text{CO})_8]^{2-}$ and $[\text{Cr}(\text{CO})_8]^{2+}$ all with O_h symmetries are optimized under M06-2X/def2tzvpp theoretical level to obtain their singlet electronic ground states. All of them are adopting the same valence electron configurations of $a_{1g}^2 t_{1u}^6 t_{2g}^6 a_{2u}^2 e_g^4$. Each orbital contributed by both ligand and metal can be directly demonstrated by manipulations with structural subunits of Metal-C bonds, which is confirmed by natural bonding orbital (NBO) analysis by using the adaptive natural density partitioning (AdNDP) method²². The AdNDP analyses are showed in Fig. 2a.	The 20-electron model for $M^q(\text{CO})_8$ ($M^q = \text{Ca}^{2-}, \text{Sc}, \text{Ti}, \text{V}^+, \text{Cr}^{2+}$) To reveal the mystery of the state of the ligand-only orbital (a_{2u} orbital), we propose to explain the bonding structures of $M^q(\text{CO})_8$ ($M^q = \text{Ca}^{2-}, \text{Sc}, \text{Ti}, \text{V}^+, \text{Cr}^{2+}$) as the singlet '20e-superatom' models based on the comparison in Fig. 1. The typical molecules selected as $[\text{Ca}(\text{CO})_8]^{2-}$, $[\text{Sc}(\text{CO})_8]^+$, $\text{Ti}(\text{CO})_8$, $[\text{V}(\text{CO})_8]^+$ and $[\text{Cr}(\text{CO})_8]^{2+}$ all with O_h symmetries are optimized under M06-2X/def2tzvpp theoretical level to obtain their singlet electronic ground states. All of them are adopting the same valence electron configurations of $a_{1g}^2 t_{1u}^6 t_{2g}^6 a_{2u}^2 e_g^4$. Each orbital contributed by both ligand and center metal can be directly demonstrated by manipulations with structural subunits of Metal-C bonds, which is confirmed by natural bonding orbital (NBO) analysis by using the adaptive natural density partitioning (AdNDP)

As for $\text{Ti}(\text{CO})_8$ (Fig. 2a), the HOMO-LUMO gap of $\text{Ti}(\text{CO})_8$ is 5.47 eV under the same theoretical level of BP86/def2tzvpp. As for $[\text{Ca}(\text{CO})_8]^{2-}$ and $[\text{Cr}(\text{CO})_8]^{2+}$, the HOMO-LUMO gaps are 3.12 eV and 9.12 eV, both of which indicate they are stable compounds. All the O atoms and carbonyl ligands appear the same configurations, which including 8 LPs of electrons localized on O atoms, 8 σ_{CO} and 16 π_{CO} bonds between C and O atoms. On the basis of AdNDP results²², the occupancy numbers (ONs) of the orbitals of the ligands (in Fig. 2a) are very close to the ideal value 2.0 e , suggesting reasonable valence properties.	method²². The AdNDP analyses are showed in Fig. 2a. As for $\text{Ti}(\text{CO})_8$ (Fig. 2a), the HOMO-LUMO gap of $\text{Ti}(\text{CO})_8$ is 5.47 eV under the same theoretical level of M062x/def2tzvpp. As for $[\text{Ca}(\text{CO})_8]^{2-}$, $[\text{Sc}(\text{CO})_8]^+$, $[\text{V}(\text{CO})_8]^+$ and $[\text{Cr}(\text{CO})_8]^{2+}$, the HOMO-LUMO gaps are 3.12 eV, 4.28 eV, 6.94 eV and 9.12 eV, respectively. There are no virtual frequencies in all the five $\text{M}^q(\text{CO})_8$ ($\text{M}^q = \text{Ca}^{2-}$, Sc^+, Ti, V^+ or Cr^{2+}) complex or ions. The vibrational frequencies of five $\text{M}^q(\text{CO})_8$ are listed in the Supporting information (SI-Table S1). Because we do not find the stable conformation of $[\text{Ca}(\text{CO})_7]^{2-}$, we only calculate the corresponding dissociation energies of $\text{M}^q(\text{CO})_8 \rightarrow \text{M}^q(\text{CO})_7 + \text{CO}$ for $[\text{Sc}(\text{CO})_8]^+$, $[\text{Ti}(\text{CO})_8]$, $[\text{V}(\text{CO})_8]^+$ and $[\text{Cr}(\text{CO})_8]^{2+}$ (The results are listed in SI-Table S2). All the results are indicating that $\text{M}^q(\text{CO})_8$ complex/ions are the minimums on the PESs. All the O atoms and carbonyl ligands appear the same configurations, which including 8 LPs of electrons localized on O atoms, 8 σ_{CO} and 16 π_{CO} bonds between C and O atoms. On the basis of AdNDP results²², the occupancy numbers (ONs) of the orbitals of the ligands (in Fig. 2a) are very close to the ideal value 2.0 e , suggesting the reasonable valence properties.
All the ONs are closed to the ideal value of 2.0 e .	All the ONs are closed to the ideal value of 2.0 e in Table 1. We deal with the other four $\text{M}^q(\text{CO})_8$ ions ($\text{M}^q = \text{Ca}^{2-}$, Sc^+, V^+ or Cr^{2+}) in the exactly same pathway to obtain the same configuration of $1\text{S}^2 1\text{P}^6 1\text{D}^{10} 1\text{F}^2$ based on the Jellium model and similar results from AdNDP method.
Page 4	
In order to make clear whether the a_{2u} orbital in $[\text{M}(\text{CO})_8]^{n+/n-}$ participates in the formation of eight coordinative bonds, we apply AdNDP method to compare the ONs of eight Ti-C orbitals of $\text{Ti}(\text{CO})_8$ with and without the participation of a_{2u} orbital (showed in Fig. 2b and 2c).	In order to make clear whether the a_{2u} orbital in any $\text{M}^q(\text{CO})_8$ ($\text{M}^q = \text{Ca}^{2-}$, Sc^+, Ti, V^+ or Cr^{2+}) participates in the formation of eight coordinative bonds, we apply AdNDP method to compare the ONs of eight Ti-C orbitals of $\text{Ti}(\text{CO})_8$ with and without the participation of a_{2u} orbital (showed in Fig. 2b and 2c).
For another aspect, the diminution of ONs from 1.94 e to 1.73 e suggests around 1/8 components (Table 1) of the coordinative orbitals have been deducted, which is the contribution of the a_{2u} orbital in the coordination. After dealing with a_{2u} orbitals of $[\text{Ca}(\text{CO})_8]^{2-}$ and $[\text{Cr}(\text{CO})_8]^{2+}$ (Fig. 2b and 2c) with the same method, we also obtain the same results on the basis of the decrease of ONs of Metal-C bonds and the same contributions of metal, respectively. The diminutions of ONs from 1.96 (1.96) to 1.72 (1.73) of $[\text{Ca}(\text{CO})_8]^{2-}$ ($[\text{Cr}(\text{CO})_8]^{2+}$) suggest the both the a_{2u} orbitals contribute around 1/8 components in the coordinative orbital (Table 1).	For another aspect, the diminution of ONs from 1.94 e to 1.71 e suggests around 1/8 components (Table 2) of the coordinative orbitals have been deducted, which is the contribution of the a_{2u} orbital in the coordination. After dealing with a_{2u} orbitals of the other four $\text{M}^q(\text{CO})_8$ ions ($\text{M}^q = \text{Ca}^{2-}$, Sc^+, V^+ or Cr^{2+}) (Fig. 2d) with the same methods, we also obtain the similar results. The diminutions of ONs from 1.96 (1.94/1.95/1.96) to 1.72 (1.71/1.72/1.73) of $[\text{Ca}(\text{CO})_8]^{2-}$ ($[\text{Sc}(\text{CO})_8]^+ / [\text{V}(\text{CO})_8]^+ / [\text{Cr}(\text{CO})_8]^{2+}$) suggest the both the a_{2u} orbitals contribute around 1/8 components in the coordinative orbital (Table 2).
Page 5	
Therefore, it can be concluded that the bonding orbitals occupy around 7/8 components (non-bonding orbital contribute 1/8 components) in the coordination. So in conclusion, the octacarbonyl metal complex ($\text{M}(\text{CO})_8$), it is more reasonable to view the ligand-only a_{2u} orbital as a non-bonding orbital contributes to form the eight coordinative orbitals ($a_{1g}^2 t_{1u}^6 e_g^6 a_{2u}^2$), where each coordinative bonds contains 1/8 non-bonding orbital and 7/8 bonding orbital as showed in Fig. 3.	Therefore, it can be concluded that the bonding orbitals occupy around 7/8 components (non-bonding orbital contribute 1/8 components) in the coordination. The similar comparison for the other three $\text{M}^q(\text{CO})_8$ ions ($\text{M}^q = \text{Sc}^+$, V^+ or Cr^{2+}) are listed in the supporting information (SI-table S3), where we can obtain the similar conclusions. So in the octacarbonyl metal complex or ions, $\text{M}^q(\text{CO})_8$ ($\text{M}^q = \text{Ca}^{2-}$, Sc^+, Ti, V^+, Cr^{2+} or Ba^{2+}), it is more reasonable to view the ligand-only a_{2u} orbital as a non-bonding orbital contributes to form the eight coordinative orbitals

	$(a_{1g}^2 t_{1u}^6 e_g^6 a_{2u}^2)$, where each coordinative bonds contains 1/8 non-bonding orbital and 7/8 bonding orbital as showed in Fig. 3.
This research demonstrates the Jellium model is a powerful analysis tool not only for metal clusters. The a_{2u} orbital is indeed a ligand-only orbital but contribute as 1/8 components in each coordinative bonds in $M(CO)_8$ compounds (or ions). Therefore, 20-e model (Jellium model) is the most appropriate explanation in understanding the valence bonding structure of $M(CO)_8$ complex.	This research demonstrates the Jellium model is a powerful analysis tool not only for metal clusters, but also for the high symmetric octacarbonyl metal complex or ions. The a_{2u} orbital is indeed a ligand-only orbital but contributing as 1/8 components in each coordinative bonds in $M^q(CO)_8$ complex or ions ($M^q=Ca^{2-}, Sc^-, Ti, V^+, Cr^{2+}$ or Ba^{2-}). Therefore, 20-e model (Jellium model) is the most appropriate explanation in understanding the valence bonding structures of $M(CO)_8$ or $TM(CO)_8$ complexes and ions.

Reviewer #3 (Remarks to the Author):

Comments:

In this paper, the authors chose a very interesting topic. The bonding nature of $M(\text{CO})_8$ ($M = \text{Ca}, \text{Sr}, \text{or Ba}$) has been thoroughly discussed in previous reports [Science 361, 912-916 (2018); Science 365, DOI: 10.1126/science.aay5021 (2019); Science 365, DOI: 10.1126/science.aay2355 (2019)]. It is innovative for the authors to use the jellium model to explain the valence bonding structure of $M(\text{CO})_8$ ($M = \text{Ca}, \text{Sr}, \text{or Ba}$) complexes. However, I think this classical jellium model is only adequate to metallic clusters. For example, the t_{2g} orbitals of $M(\text{CO})_8$ in Fig. 1 are assigned to 1D superatomic orbitals are not reasonable since they are more like F orbitals in the jellium model. The same cases are also found in Fig. 2a, wherein the t_{1u} are considered as 1P orbitals are ridiculous. Hence, I think this paper is not appropriate to be published in this journal. If the authors want to publish this work in another theoretical journal, the following mistakes should be seriously modified:

Jellium model is a quantum mechanical model of interacting electrons in a solid where the positive charges are assumed to be uniformly distributed in space whence the electron density is a uniform quantity as well in space, which usually applied in uniform electron gas or homogeneous electron gas model. But in our systems, all the octacarbonyl metal complexes are with O_h symmetries, where the octa-coordination fields contributed by the positive center metal and eight ligands can be approximately viewed as homogeneous spherical fields. Furthermore, this is demonstrated by the similarity of the metal cluster $[\text{BaBe}_8]^{2-}$ and a designed octacarbonyl metal dianion $[\text{Ba}(\text{CO})_8]^{2-}$. The detailed analysis has been discussed in the manuscript:

The comparison of the molecular orbitals of $[\text{Ba}(\text{CO})_8]^{2-}$ and $[\text{Ba}(\text{Be})_8]^{2-}$ are showed in Fig. 1. Although $[\text{Ba}(\text{CO})_8]^{2-}$ is not a metal cluster, its configuration can be similarly described as $|1S^2|1P^6|1D^6|1F^2|1D^4|$ based on the Jellium model. Furthermore, a_{1g} , t_{1u} and t_{2g} orbitals of $[\text{Ba}(\text{CO})_8]^{2-}$ are corresponding to the 1S, 1P orbitals and $1D_{xy}/1D_{yz}/1D_{xz}$ super orbitals. The e_g MOs are corresponding to the $1D_{x^2-y^2}$ and $1D_{z^2}$ orbitals as two π -type orbitals, corresponding to the two $d \rightarrow \pi^*$ backdonation bonds donated from 5d AOs of Ba ($5d_{x^2-y^2}$ and $5d_{z^2}$ AOs) to the 2p AOs of eight CO groups. The ligand-only a_{2u} orbital composed of eight carbonyl groups is also defined as the 1F super orbital based on the diagram in Fig. 1. The similarity of $[\text{Ba}(\text{CO})_8]^{2-}$ and $[\text{BaBe}_8]^{2-}$ indicate we can understand the valence orbitals of $[\text{Ba}(\text{CO})_8]^{2-}$ as a '20e-superatom' based on Jellium model.

As for the suggestions, we add the explanation about the reason why we assume to apply Jellium model for $M(\text{CO})_8$ complexes or ions in the revised manuscript:

For another aspect, the octacarbonyl metal complex or ion is with O_h symmetries, where the octa-coordination field contributed by the positive center metal and eight ligands can be approximately viewed as a homogeneous spherical field or a Jellium model^{12,13}. Therefore, we design a series of O_h -symmetric and singlet-state $M^q(\text{CO})_8$ ($M^q = \text{Ca}^{2-}, \text{Sc}^-, \text{Ti}, \text{V}^+, \text{Cr}^{2+}$ or Ba^{2-}) complex/ions theoretically to learn their bonding structures.

As for the t_{2g} orbitals of $M(\text{CO})_8$, we have modified the isovalues for showing a more clear D-type superorbitals. The comparison of the old version and the new version has been showed as Fig.1 below, where we can see the 3-degenerate d orbitals localized on the metal clearly.

Fig.1 The comparison of t_{2g} orbitals of $M(CO)_8$ in the old version and the new version

As for the t_{1u} orbitals in Fig.2, we tuned the isovalue of the density surfaces and showed the AOs of metal in the AdNDP analysis, which can be clearly found the t_{1u} orbitals are mainly contributed from the 3-degenerate p -type orbitals of the metal.

Fig 2 AdNDP analysis of $Ti(CO)_8$

On the basis of the suggestions, the pictures have been also updated in Fig.1 and Fig.2 of the revised manuscript, which also showed as below:

Fig. 1 The molecular orbitals listed on the basis of Jellium model of singlet $[\text{Ba}(\text{Be})_8]^{2-}$ (left) and singlet $[\text{Ba}(\text{CO})_8]^{2-}$ (right).

Fig. 2 (a) AdNDP analysis of $\text{M}^q(\text{CO})_8$ ($\text{M}^q = \text{Ca}^{2+}$, Sc^+ , Ti , V^+ or Cr^{2+}) complex/ions, (b) The detailed AdNDP analysis of (|ON|s) of $\text{M}^q(\text{CO})_8$ complex/ions with the participation of a_{2u} orbital and (c) without the participation of a_{2u} orbital. (d) The occupied number (|ON|s) of $\text{M}^q(\text{CO})_8$ complex/ions with/without the participation of a_{2u} orbital.

1) On Pg 2, line 13(left column) “alkali metal” should be “alkaline earth metal”;

We have modified “alkali metal” should be “alkaline earth metal” as the suggestions:

The recent experimentally produced alkaline earth metal complexes $\text{M}(\text{CO})_8$ ($\text{M} = \text{Ca}$, Sr and Ba) are satisfying the DCD model and 18-electron rule,^{6,7} including 8 degenerate $\sigma_{\text{CO} \rightarrow \text{M}}$ coordinative bonds and 2 (n-1)d $\rightarrow \pi^*$ backdonation bonds from alkali earth metal to the ligands.

2) on Pg 3, line 2 of the second paragraph, the computational method “BP86/def2tzvpp” should be “M06-2X/def2tzvpp” according to their supporting information.

Yes. It is an error in the previous manuscript. Now we have modified in the revised paper: As for $\text{Ti}(\text{CO})_8$ (Fig. 2a), the HOMO-LUMO gap of $\text{Ti}(\text{CO})_8$ is 5.47 eV under the same theoretical level of M062x/def2tzvpp.

3) Thus, the authors should check the main text carefully again.

We have rewritten most part of the manuscript based on all the comments. Here are the details, where all the modifications are highlighted in the table

Page 1	
The recently reported octacarbonyl metal complexes $\text{M}(\text{CO})_8$ appear very interesting bonding structures. In these compounds, there are 8 coordinative bonds ($\text{CO} \rightarrow \text{Metal}$) and 2 backdonation bonds ($d \rightarrow \pi^*$) based on the valence bond (VB) theory. However, there are only seven valence orbitals (d^3sp^3 hybridization with the bond order of 7) participating in forming σ -type bonds. This is an interesting but a common phenomenon that the bond order is 7 but with accommodating 8 LPs of ligands in forming octa-coordinated complex. Here we demonstrate Jellium model is also a useful tool in understanding the electronic structure of $\text{M}(\text{CO})_8$ not only for metal clusters. By applying Jellium model, the magic stability of $\text{M}(\text{CO})_8$ complex is because the structure is satisfying the '20-electron' rule (20-e is the magic number of the Jellium model) by comparing $\text{M}(\text{CO})_8$ and metal clusters of BaBe_8 analogically. As for the 8 coordinative bonds, each coordinative bond contains 1/8 components of the non-bonding orbital (or $ 1F^2\rangle$ orbital in Jellium model) in $\text{M}(\text{CO})_8$ complexes.	The recently reported octacarbonyl metal complexes $\text{M}(\text{CO})_8$ ($\text{M} = \text{Ca}, \text{Sr}$ and Ba) appear very interesting bonding structures. In these compounds, there are 8 coordinative bonds ($\text{CO} \rightarrow \text{Metal}$) and 2 backdonation bonds ($d \rightarrow \pi^*$) based on the valence bond (VB) theory. However, there are only seven valence orbitals (d^3sp^3 hybridization with the bond order of 7) participating in forming σ -type bonds. This is an interesting but a common phenomenon that the bond order is 7 but with accommodating 8 lone pairs (LPs) of ligands in forming octa-coordinated complexes/ions. By comparing $[\text{Ba}(\text{CO})_8]^{2-}$ and metal clusters of $[\text{BaBe}_8]^{2-}$ analogically, we demonstrate Jellium model is also a useful tool in understanding the electronic structures of $\text{M}^q(\text{CO})_8$ ($\text{M}^q = \text{Ca}^{2-}, \text{Sc}^-, \text{Ti}, \text{V}^+, \text{Cr}^{2+}$ or Ba^{2-}) rather than only for metal clusters. By applying Jellium model, the magic stabilities of such $\text{M}^q(\text{CO})_8$ complex or ions are because the structure is satisfying the '20-electron' rule (20-e is the magic number of the Jellium model). As for the 8 coordinative bonds, each coordinative bond contains 1/8 components of the non-bonding orbital (or $ 1F^2\rangle$ orbital in Jellium model) in $\text{M}^q(\text{CO})_8$ complexes or ions
Page 2	
The 18-electron rule is a very classic tool for us to understand the structures of a large amount of metal complexes. ¹ Especially for the homoleptic carbonyl metal complexes, it is very successful to apply Dewar-Chatt-Duncanson (DCD) model ² and 18-electron rule to explain the interactions between carbon monoxide and metals, such as the seven-coordinate carbonyl complexes $[\text{M}(\text{CO})_7]^-$ ($\text{M} = \text{V}^+, \text{Nb}^+$ and Ta^+) and the eight-coordinated carbonyl complexes $[\text{M}(\text{CO})_8]^+$ ($\text{M} = \text{Sc}^+, \text{Y}^+$ and La^+). ³⁻⁵ The recent experimentally produced alkaline earth metal complexes $\text{M}(\text{CO})_8$ ($\text{M} = \text{Ca}, \text{Sr}$ and Ba) are satisfying the DCD model and 18-electron rule, ^{6,7} including 8 degenerate $\sigma_{\text{CO}_3\text{M}}$ coordinative bonds and 2 $(n-1)d \rightarrow \pi^*$ backdonation bonds from alkali metal to the ligands. ⁸	The 18-electron rule is a very classic tool for us to understand the structures of a large amount of transition metal complexes. ¹ Especially for the homoleptic carbonyl metal complexes, it is very successful to apply Dewar-Chatt-Duncanson (DCD) model ² and 18-electron rule to explain the interactions between carbon monoxide and transition metals, such as the seven-coordinate carbonyl cations $[\text{TM}(\text{CO})_7]^+$ ($\text{TM} = \text{V}^+, \text{Nb}^+$ and Ta^+) and the eight-coordinated carbonyl anions $[\text{TM}(\text{CO})_8]^-$ ($\text{TM} = \text{Sc}^-, \text{Y}^-$ and La^-). ³⁻⁵ The recent experimentally produced alkaline earth metal complexes $\text{M}(\text{CO})_8$ ($\text{M} = \text{Ca}, \text{Sr}$ and Ba) are satisfying the DCD model and 18-electron rule, ^{6,7} including 8 degenerate $\sigma_{\text{CO}_3\text{TM}}$ coordinative bonds and 2 $(n-1)d \rightarrow \pi^*$ backdonation bonds from alkali earth metal to the ligands. ⁸
There is an interesting inconsistency in understanding the electronic structures of octacarbonyl complexes $\text{M}(\text{CO})_8$ ($\text{M} = \text{Ca}, \text{Sr}$ and Ba) or cations $[\text{M}(\text{CO})_8]^+$ ($\text{M} = \text{Sc}^+, \text{Y}^+$ and La^+) on the basis of different chemical bonding theories. It should be noted that all the structures adopt O_h symmetry. Therefore, on the basis of valence bond (VB) theory ⁹ , there are eight degenerate coordination bonds and two π -backdonation bonds between metal and ligands in the series $\text{M}(\text{CO})_8$	There is an interesting inconsistency in understanding the electronic structures of octacarbonyl complexes $\text{M}(\text{CO})_8$ ($\text{M} = \text{Ca}, \text{Sr}$ and Ba) or cations $[\text{TM}(\text{CO})_8]^+$ ($\text{TM} = \text{Sc}^+, \text{Y}^+$ and La^+) on the basis of different chemical bonding theories. It should be noted that all the structures adopt O_h symmetry. Therefore, on the basis of valence bond (VB) theory ⁹ , there are eight degenerate coordination bonds and two π -backdonation bonds between metal and ligands in the series $\text{M}^q(\text{CO})_8$

complexes, where eight carbonyl groups as Lewis base provide eight lone pairs (LPs) of electrons to the metal. However, based on the hybrid orbital (HO) theory¹⁰, only seven empty degenerate orbitals can be provided by the metal by adopting d^3sp^3 hybridization. Therefore, the bond orders are inconsistent on the basis of the two classic theories. It is difficult to arrange the 16 LPs in forming an eight-coordinated complex with the bond order of 7. It has been hypothesized that the eight σ-type orbitals are contributed by both the ligands and the metal, where the metal possibly adopts a higher-level hybridization (such as the f-type polarization³) for bonding with the eighth carbonyl groups. It is a reasonable viewpoint to understand the inconsistency of VB and HO theories, where the bond orders are both 8.	complexes or ions, where eight carbonyl groups provide eight lone pairs (LPs) of electrons to the center metal. However, based on the hybrid orbital (HO) theory¹⁰, only seven empty degenerate orbitals can be provided by the d^3sp^3 hybridized metal. Therefore, the bond orders are inconsistent on the basis of the two classic theories. It is difficult to arrange the 16 LPs in forming an eight-coordinated complex with the bond order of 7. As for $\text{TM}(\text{CO})_8$, it has been hypothesized that the eight σ-type orbitals are contributed by both the ligands and the transition metal, where the transition metal possibly adopts a higher-level hybridization (such as the f-type polarization³) for bonding with the eighth carbonyl groups. It is a reasonable viewpoint to understand the inconsistency of VB and HO theories, where the bond orders are both equal to 8.
Additionally, Zhou's research pointed out that the molecules adopt O_h symmetry with the valence electron configuration of $a_{1g}^2t_{1u}^6t_{2g}^6a_{2u}^2e_g^2$, where the a_{2u} orbital is explained as a ligand-only orbital, which is satisfying the 18-electron rule perfectly^{3,8}. The electrons distributed in a_{2u} orbital is stabilized by the field effect of the metal on the ligand cage.^{3,11} Besides the ligand-only a_{2u} orbital, the other MOs ($a_{1g} + 3t_{1u} + 3t_{2g} + 2e_g$) including 7 σ-donation bonds ($a_{1g} + 3t_{1u} + 3t_{2g}$) and 2 π-backdonation bonds ($2e_g$) can accommodate 18 valence electrons of $\text{M}(\text{CO})_8$ complex perfectly,⁸ which is also quite reasonable to understand the bonding structure of $\text{M}(\text{CO})_8$ compounds or ions. However, it is attractive for us to understand the state of the ligand-only orbital (a_{2u} orbital) essentially. $\text{M}(\text{CO})_8$ complex is a 20-e closed-shell molecule based on the configuration³, which is exactly satisfying the magic stability in Jellium model^{12,13}. Is it possible the ligand-only orbital as a component participating in the eight coordinative bonds to form a 20-electron configuration for its valence electrons?	Additionally, Zhou's research pointed out that $\text{M}(\text{CO})_8$ complexes or ions adopt O_h symmetry with the valence electron configuration of $a_{1g}^2t_{1u}^6t_{2g}^6a_{2u}^2e_g^2$, where the a_{2u} orbital is explained as the ligand-only orbital, which is satisfying the 18-electron rule perfectly^{3,8}. The electrons in a_{2u} orbital are stabilized by the field effect of the metal on the ligand cage.^{3,11} Besides the ligand-only a_{2u} orbital, the other 9 MOs ($a_{1g} + 3t_{1u} + 3t_{2g} + 2e_g$) including 7 σ-donation bonds ($a_{1g} + 3t_{1u} + 3t_{2g}$) and 2 π-backdonation bonds ($2e_g$) can accommodate 18 valence electrons of $\text{M}(\text{CO})_8$ complex perfectly,⁸ which is also quite reasonable to understand the bonding structure of $\text{M}(\text{CO})_8$ complexes or ions. On the basis of the experimental studies, all the 10 valence orbitals ($a_{1g} + 3t_{1u} + 3t_{2g} + a_{2u} + 2e_g$) can be fulfilled with maximum 20 electrons as the singlet states for the O_h-symmetric octacarbonyl metal complexes or ions. Therefore, it is attractive for us to understand the state of the ligand-only orbital (a_{2u} orbital) essentially. For another aspect, the octacarbonyl metal complex or ion is with O_h symmetries, where the octa-coordination field contributed by the positive center metal and eight ligands can be approximately viewed as a homogeneous spherical field or a Jellium model^{12,13}. Therefore, we design a series of O_h-symmetric and singlet-state $\text{M}^q(\text{CO})_8$ ($\text{M}^q = \text{Ca}^{2+}, \text{Sc}^+, \text{Ti}, \text{V}^+, \text{Cr}^{2+}$ or Ba^{2+}) complex/ions theoretically to learn their bonding structures. Such $\text{M}^q(\text{CO})_8$ complex/ions are 20-e closed-shell molecules based on the configuration³, which is exactly satisfying the magic stability of Jellium model^{12,13}. We are curious that whether the ligand-only a_{2u} orbital as a component participate in the eight coordinative bonds to form a 20-electron configuration for its valence electrons?
As for the analogical octa-coordinative complex, the O_h-symmetric $[\text{BaBe}_8]^{2-}$ are optimized at M06-2X/def2tzvpp level of theory^{19,20} in Gaussian 09²¹ to obtain the singlet electronic ground state with the valence electron configuration $a_{1g}^2t_{1u}^6t_{2g}^6a_{2u}^2e_g^4$ with the HOMO-LUMO gap of 2.77 eV. The electrons fulfil the valence orbital following the configuration of $1S^2 1P^6 1D^6 1F^2 1D^4$, where D orbitals are split into two groups as ($1D_{xy}, 1D_{yz}$ and $1D_{xz}$) and ($1D_{x^2-y^2}$ and $1D_{z^2}$). It should be noticed that the a_{2u} orbital adopts the f symmetry as showed in Fig. 1. In the configuration, the number of valence electrons just equals to the	As for the analogical octa-coordinative complex, the O_h-symmetric $[\text{BaBe}_8]^{2-}$ are optimized at M06-2X/def2tzvpp level of theory^{19,20} in Gaussian 09²¹ to obtain the singlet electronic ground state. The valence electron configuration of $[\text{BaBe}_8]^{2-}$ is $a_{1g}^2t_{1u}^6t_{2g}^6a_{2u}^2e_g^4$ with the HOMO-LUMO gap of 2.77 eV. Based on the Jellium model, the electrons fulfil the 10 valence orbitals following the configuration of $1S^2 1P^6 1D^6 1F^2 1D^4$, where D orbitals are split into two groups as ($1D_{xy}, 1D_{yz}$ and $1D_{xz}$) and ($1D_{x^2-y^2}$ and $1D_{z^2}$). It should be noticed that the a_{2u} orbital is an 1F orbital rather than 2S orbital, where the energy of 1F orbital is in

magic number '20' appearing the magical stability.	the middle of the two groups 1D orbitals. Based on the calculation, a_{2u} orbital adopts the f symmetry as showed in Fig. 1. Furthermore, there is a cubic field in all the $M^q(\text{CO})_8$ complexes or ions, which affect the 7-degenerate 1F orbitals in the coordination, in which the a_{2u}-symmetric 1F_{xyz} orbital matches the orbital symmetry in the cubic field of coordination. Therefore, the energy of 1F_{xyz} orbital is lower than 2S orbital caused by the splits of F orbitals (Figure S1 in the supporting information). The sequences of energy levels of the orbitals are the results of the splitting of 1D and 1F orbitals. In the configuration, the number of valence electrons just equals to the magic number '20' appearing the magical stability.
Page 3	
The comparison of the molecular orbitals of $[\text{Ba}(\text{CO})_8]^{2-}$ and $[\text{Ba}(\text{Be})_8]^{2-}$ are showed in Fig. 1. Although $[\text{Ba}(\text{CO})_8]^{2-}$ is not a metal cluster, its configuration can be similarly described as jellium model as $[1S^2]1P^6]1D^6]1F^2]1D^4]$. As for $[\text{Ba}(\text{CO})_8]^{2-}$, a_{1g}, t_{1u} and t_{2g} orbitals are corresponding to the 1S, 1P orbitals and 1D_{xy}/1D_{yz}/1D_{xz} super orbitals in Jellium model. The e_g MOs are corresponding to the 1D_{x²-y²} and 1D_{z²} orbitals as two π-type orbitals (or 5d_{x²-y²} and 5d_{z²} AOs of barium), which are the two $d \rightarrow \pi^*$ bonds donated from 5d AOs of Ba to the 2p AOs of eight CO groups. The ligand-only a_{2u} orbital composed of eight carbonyl groups is 1F super orbital based on the diagram in Fig. 1. The similarity of $[\text{Ba}(\text{CO})_8]^{2-}$ and $[\text{BaBe}_8]^{2-}$ indicate we can understand the complex as a '20e-superatom' based on Jellium model (20 is a magic number suggesting the structural stability). For another aspect, we apply MO theory to understand the eight coordinative bonds in $[\text{Ba}(\text{CO})_8]^{2-}$. There should be 15 MOs combined by 7 valence AOs of Ba atom (d^3sp^3) and 8 AOs of LPs donated from 8 (CO) groups in forming eight $\sigma(\text{Ba-C})$ bonds based on MO theory. Therefore, the 15 MOs should be composed as 7 bonding, 1 non-bonding, and 7 anti-bonding orbitals filled by 16 electrons successively. In the 15 MOs, the 7 bonding orbitals are corresponding to the a_{1g}, t_{1u} and t_{2g} orbitals. The non-bonding orbital is exactly corresponding to the ligand-only a_{2u} orbital.	The comparison of the molecular orbitals of $[\text{Ba}(\text{CO})_8]^{2-}$ and $[\text{Ba}(\text{Be})_8]^{2-}$ are showed in Fig. 1. Although $[\text{Ba}(\text{CO})_8]^{2-}$ is not a metal cluster, its configuration can be similarly described as $[1S^2]1P^6]1D^6]1F^2]1D^4]$ based on the Jellium model. Furthermore, a_{1g}, t_{1u} and t_{2g} orbitals of $[\text{Ba}(\text{CO})_8]^{2-}$ are corresponding to the 1S, 1P orbitals and 1D_{xy}/1D_{yz}/1D_{xz} super orbitals. The e_g MOs are corresponding to the 1D_{x²-y²} and 1D_{z²} orbitals as two π-type orbitals, corresponding to the two $d \rightarrow \pi^*$ bonds donated from 5d AOs of Ba (5d_{x²-y²} and 5d_{z²} AOs) to the 2p AOs of eight CO groups. The ligand-only a_{2u} orbital composed of eight carbonyl groups is also defined as the 1F super orbital based on the diagram in Fig. 1. The similarity of $[\text{Ba}(\text{CO})_8]^{2-}$ and $[\text{BaBe}_8]^{2-}$ indicate we can understand the valence orbitals of $[\text{Ba}(\text{CO})_8]^{2-}$ as a '20e-superatom' based on Jellium model. For another aspect, we apply MO theory to understand the eight σ-type coordinative bonds in $[\text{Ba}(\text{CO})_8]^{2-}$. There should be 15 MOs contributed by 7 valence AOs of Ba atom (d^3sp^3) and 8 AOs of LPs donated from 8 (CO) groups in forming eight $\sigma(\text{Ba-C})$ bonds based on MO theory. Therefore, the 15 MOs should be composed as 7 bonding, 1 non-bonding, and 7 anti-bonding orbitals filled by 16 electrons successively. In the 15 MOs, the 7 bonding orbitals are corresponding to the a_{1g}, t_{1u} and t_{2g} orbitals. The non-bonding orbital is exactly corresponding to the ligand-only a_{2u} orbital. This is also consistent with the sequence of energies of orbital levels, where the energy of 1F_{xyz} orbital as a non-bonding orbital is between 3-degenerate 1D orbitals (1D_{xy}, xz, yz) and 2-degenerate 1D orbitals (1D_{x²-y²} and 1D_{z²}).
Page 4	
The 20-electron model for $M(\text{CO})_8$ To reveal the mystery of the state of the ligand-only orbital (a_{2u} orbital), we propose to explain the bonding structure of $[\text{M}(\text{CO})_8]$ complexes as a singlet '20e-superatom' model based on the comparisons in Fig. 1. The typical molecules selected as $\text{Ti}(\text{CO})_8$, $[\text{Ca}(\text{CO})_8]^{2-}$ and $[\text{Cr}(\text{CO})_8]^{2+}$ all with O_h symmetries are optimized under M06-2X/def2tzvpp theoretical level to obtain their singlet electronic ground states. All of them are adopting the same valence electron configurations of $a_{1g}^2 t_{1u}^6 t_{2g}^6 a_{2u}^2 e_g^4$. Each orbital contributed by both ligand and metal can be directly demonstrated by manipulations with structural subunits of Metal-C bonds, which is confirmed by natural bonding orbital (NBO) analysis by using the adaptive natural density	The 20-electron model for $M^q(\text{CO})_8$ ($M^q=\text{Ca}^{2+}$, Sc^+, Ti, V^+ or Cr^{2+}) To reveal the mystery of the state of the ligand-only orbital (a_{2u} orbital), we propose to explain the bonding structures of $M^q(\text{CO})_8$ ($M^q=\text{Ca}^{2+}$, Sc^+, Ti, V^+ or Cr^{2+}) as the singlet '20e-superatom' models based on the comparison in Fig. 1. The typical molecules selected as $[\text{Ca}(\text{CO})_8]^{2-}$, $[\text{Sc}(\text{CO})_8]^+$, $\text{Ti}(\text{CO})_8$, $[\text{V}(\text{CO})_8]^+$ and $[\text{Cr}(\text{CO})_8]^{2+}$ all with O_h symmetries are optimized under M06-2X/def2tzvpp theoretical level to obtain their singlet electronic ground states. All of them are adopting the same valence electron configurations of $a_{1g}^2 t_{1u}^6 t_{2g}^6 a_{2u}^2 e_g^4$. Each orbital contributed by both ligand and center metal can be directly demonstrated by manipulations with structural subunits of Metal-C bonds, which is confirmed by natural bonding

partitioning (AdNDP) method²². The AdNDP analyses are showed in Fig. 2a. As for $\text{Ti}(\text{CO})_8$ (Fig. 2a), the HOMO-LUMO gap of $\text{Ti}(\text{CO})_8$ is 5.47 eV under the same theoretical level of BP86/def2tzvpp. As for $[\text{Ca}(\text{CO})_8]^{2-}$ and $[\text{Cr}(\text{CO})_8]^{2+}$, the HOMO-LUMO gaps are 3.12 eV and 9.12 eV, both of which indicate they are stable compounds. All the O atoms and carbonyl ligands appear the same configurations, which including 8 LPs of electrons localized on O atoms, 8 σ_{CO} and 16 π_{CO} bonds between C and O atoms. On the basis of AdNDP results²², the occupancy numbers (ONs) of the orbitals of the ligands (in Fig. 2a) are very close to the ideal value 2.0 e , suggesting reasonable valence properties.	orbital (NBO) analysis by using the adaptive natural density partitioning (AdNDP) method²². The AdNDP analyses are showed in Fig. 2a. As for $\text{Ti}(\text{CO})_8$ (Fig. 2a), the HOMO-LUMO gap of $\text{Ti}(\text{CO})_8$ is 5.47 eV under the same theoretical level of M062x/def2tzvpp. As for $[\text{Ca}(\text{CO})_8]^{2-}$, $[\text{Sc}(\text{CO})_8]$, $[\text{V}(\text{CO})_8]^+$ and $[\text{Cr}(\text{CO})_8]^{2+}$, the HOMO-LUMO gaps are 3.12 eV, 4.28 eV, 6.94 eV and 9.12 eV, respectively. There are no virtual frequencies in all the five $\text{M}^q(\text{CO})_8$ ($\text{M}^q=\text{Ca}^{2-}$, Sc^-, Ti, V^+ or Cr^{2+}) complex or ions. The vibrational frequencies of five $\text{M}^q(\text{CO})_8$ are listed in the Supporting information (SI-Table S1). Because we do not find the stable conformation of $[\text{Ca}(\text{CO})_7]^{2-}$, we only calculate the corresponding dissociation energies of $\text{M}^q(\text{CO})_8 \rightarrow \text{M}^q(\text{CO})_7 + \text{CO}$ for $[\text{Sc}(\text{CO})_8]$, $[\text{Ti}(\text{CO})_8]$, $[\text{V}(\text{CO})_8]^+$ and $[\text{Cr}(\text{CO})_8]^{2+}$ (The results are listed in SI-Table S2). All the results are indicating that $\text{M}^q(\text{CO})_8$ complex/ions are the minimums on the PESSs. All the O atoms and carbonyl ligands appear the same configurations, which including 8 LPs of electrons localized on O atoms, 8 σ_{CO} and 16 π_{CO} bonds between C and O atoms. On the basis of AdNDP results²², the occupancy numbers (ONs) of the orbitals of the ligands (in Fig. 2a) are very close to the ideal value 2.0 e , suggesting the reasonable valence properties.
All the ONs are closed to the ideal value of 2.0 e .	All the ONs are closed to the ideal value of 2.0 e in Table 1. We deal with the other four $\text{M}^q(\text{CO})_8$ ions ($\text{M}^q=\text{Ca}^{2-}$, Sc^-, V^+ or Cr^{2+}) in the exactly same pathway to obtain the same configuration of $1\text{S}^2 1\text{P}^6 1\text{D}^{10} 1\text{F}^2$ based on the Jellium model and similar results from AdNDP method.
Page 4	
In order to make clear whether the a_{2u} orbital in $[\text{M}(\text{CO})_8]^{n+}$ participates in the formation of eight coordinative bonds, we apply AdNDP method to compare the ONs of eight Ti-C orbitals of $\text{Ti}(\text{CO})_8$ with and without the participation of a_{2u} orbital (showed in Fig. 2b and 2c).	In order to make clear whether the a_{2u} orbital in any $\text{M}^q(\text{CO})_8$ ($\text{M}^q=\text{Ca}^{2-}$, Sc^-, Ti, V^+ or Cr^{2+}) participates in the formation of eight coordinative bonds, we apply AdNDP method to compare the ONs of eight Ti-C orbitals of $\text{Ti}(\text{CO})_8$ with and without the participation of a_{2u} orbital (showed in Fig. 2b and 2c).
For another aspect, the diminution of ONs from 1.94 e to 1.73 e suggests around 1/8 components (Table 1) of the coordinative orbitals have been deducted, which is the contribution of the a_{2u} orbital in the coordination. After dealing with a_{2u} orbitals of $[\text{Ca}(\text{CO})_8]^{2-}$ and $[\text{Cr}(\text{CO})_8]^{2+}$ (Fig. 2b and 2c) with the same method, we also obtain the same results on the basis of the decrease of ONs of Metal-C bonds and the same contributions of metal, respectively. The diminutions of ONs from 1.96 (1.96) to 1.72 (1.73) of $[\text{Ca}(\text{CO})_8]^{2-}$ ($[\text{Cr}(\text{CO})_8]^{2+}$) suggest the both the a_{2u} orbitals contribute around 1/8 components in the coordinative orbital (Table 1).	For another aspect, the diminution of ONs from 1.94 e to 1.71 e suggests around 1/8 components (Table 2) of the coordinative orbitals have been deducted, which is the contribution of the a_{2u} orbital in the coordination. After dealing with a_{2u} orbitals of the other four $\text{M}^q(\text{CO})_8$ ions ($\text{M}^q=\text{Ca}^{2-}$, Sc^-, V^+ or Cr^{2+}) (Fig. 2d) with the same methods, we also obtain the similar results. The diminutions of ONs from 1.96 (1.94/1.95/1.96) to 1.72 (1.71/1.72/1.73) of $[\text{Ca}(\text{CO})_8]^{2-}$ ($[\text{Sc}(\text{CO})_8]$/$[\text{V}(\text{CO})_8]^+$/$[\text{Cr}(\text{CO})_8]^{2+}$) suggest the both the a_{2u} orbitals contribute around 1/8 components in the coordinative orbital (Table 2).
Page 5	
Therefore, it can be concluded that the bonding orbitals occupy around 7/8 components (non-bonding orbital contribute 1/8 components) in the coordination. So in conclusion, the octacarbonyl metal complex ($\text{M}(\text{CO})_8$), it is more reasonable to view the ligand-only a_{2u} orbital as a non-bonding orbital contributes to form the eight coordinative orbitals ($a_{1g}^2 t_{1u}^6 t_{2g}^6 a_{2u}^2$), where each coordinative bonds contains 1/8 non-bonding orbital and 7/8 bonding orbital as showed in Fig. 3.	Therefore, it can be concluded that the bonding orbitals occupy around 7/8 components (non-bonding orbital contribute 1/8 components) in the coordination. The similar comparison for the other three $\text{M}^q(\text{CO})_8$ ions ($\text{M}^q=\text{Sc}^-$, V^+ or Cr^{2+}) are listed in the supporting information (SI-table S3), where we can obtain the similar conclusions. So in the octacarbonyl metal complex or ions, $\text{M}^q(\text{CO})_8$ ($\text{M}^q=\text{Ca}^{2-}$, Sc^-, Ti, V^+, Cr^{2+} or Ba^{2-}), it is more reasonable to view the ligand-only a_{2u} orbital as a

	non-bonding orbital contributes to form the eight coordinative orbitals ($a_{1g}^2 t_{1u}^6 t_{2g}^6 a_{2u}^2$), where each coordinative bonds contains 1/8 non-bonding orbital and 7/8 bonding orbital as showed in Fig. 3.
This research demonstrates the Jellium model is a powerful analysis tool not only for metal clusters. The a_{2u} orbital is indeed a ligand-only orbital but contribute as 1/8 components in each coordinative bonds in $M(CO)_8$ compounds (or ions). Therefore, 20-e model (Jellium model) is the most appropriate explanation in understanding the valence bonding structure of $M(CO)_8$ complex.	This research demonstrates the Jellium model is a powerful analysis tool not only for metal clusters, but also for the high symmetric octacarbonyl metal complex or ions. The a_{2u} orbital is indeed a ligand-only orbital but contributing as 1/8 components in each coordinative bonds in $M^q(CO)_8$ complex or ions ($M^q=Ca^{2+}$, Sc^+, Ti, V^+, Cr^{2+} or Ba^{2+}). Therefore, 20-e model (Jellium model) is the most appropriate explanation in understanding the valence bonding structures of $M(CO)_8$ or $TM(CO)_8$ complexes and ions.

Reviewers' comments:

Reviewer #1 (Remarks to the Author):

I think most of issues have been addressed, now it can be suitable for publication.

Reviewer #2 (Remarks to the Author):

The authors have successfully addressed all the points by the Reviewers. So I recommend publication of this manuscript in Communications Chemistry.

Reviewer #3 (Remarks to the Author):

By following the reviewers' comments, the quality of the revised manuscript has been greatly improved. However, my main criticism concerns the fact that these studied metal complex or ions $Mq(CO)_8$ ($Mq = Ca^{2-}, Sc-, Ti, V+, Cr^{2+}$ or Ba^{2-}) are not thermodynamically stable considering their negative dissociation energies. If these species are not stable, all the discussion in this paper becomes meaningless since the $Mq(CO)_7$ ($Mq = Ca^{2-}, Sc-, Ti, V+, Cr^{2+}$ or Ba^{2-}) with 18 electrons are more stable, and thus can be understood by the simple 18-electron rule. Hence, I think the authors should explain this point before it can be published in this journal. Moreover, there are still a lot of minor mistakes in the current version, some are listed as follows:

1. On Pg 2, line 3(left column) of the second paragraph, "cations $[TM(CO)_8]^-$ ($TM = Sc-, Y-$ and $La-$)" should be "anions $[TM(CO)_8]^-$ ($TM = Sc-, Y-$ and $La-$)"; line 8 (left column) of the second paragraph, "in the series $M(CO)_8$ complexes" should be "in the $M(CO)_8$ complexes". line 4 (right column) of the second paragraph of "Results", "with Oh symmetries" should be "with Oh symmetry".
2. On Pg 3, line 2 (left column) of the first paragraph, "showed" should be "shown"; line 12 (left column) of the first paragraph, "indicate" should be "indicates".
3. On Pg 4, line 9 (left column) of the first paragraph, "are adopting" should be "adopts" (a lot of such tense mistakes in the context); line 14 (right column) of the second paragraph, "which are exactly the same with before" should be corrected.
4. On Pg 5, line 5 (left column) of the second paragraph, "bonds" should be "bond"; line 6 (left column) of the second paragraph, "showed" should be "shown"; line 7 (left column) of the second paragraph, "orbital" should be "orbitals".

Response to the Reviewers

Reviewer #3 (Remarks to the Author):

By following the reviewers' comments, the quality of the revised manuscript has been greatly improved. However, my main criticism concerns the fact that these studied metal complex or ions $M^q(CO)_8$ ($M^q = Ca^{2-}, Sc^-, Ti, V^+, Cr^{2+}$ or Ba^{2-}) are not thermodynamically stable considering their negative dissociation energies. If these species are not stable, all the discussion in this paper becomes meaningless since the $M^q(CO)_7$ ($M^q = Ca^{2-}, Sc^-, Ti, V^+, Cr^{2+}$ or Ba^{2-}) with 18 electrons are more stable, and thus can be understood by the simple 18-electron rule. Hence, I think the authors should explain this point before it can be published in this journal.

We hope to explain this query in the following 2 aspects.

I) $M^q(CO)_7$ and $M^q(CO)_8$ systems can be both synthesized successfully

It is truly that $M^q(CO)_7$ systems are more stable than that of $M^q(CO)_8$ ($M^q = Ca^{2-}, Sc^-, Ti, V^+, Cr^{2+}$ or Ba^{2-}). However, in the experimental field, $M^q(CO)_7$ ($M^q = Sc^-, Y^-, La^-, V^+, Nb^+$ and Ta^+) and $M^q(CO)_8$ ($M^q = Ca, Sr, Ba, Sc^-, Y^-, La^-, Sc^+, Y^+$ and La^+) are both synthesized in gas phase experimentally, where $M^q(CO)_7$ ($M^q = Sc^-, Y^-, La^-, V^+, Nb^+$ and Ta^+) and $M^q(CO)_8$ ($M^q = Sc^+, Y^+$ and La^+) systems satisfy the 18-e rule very well (Angew. Chem. Int. Ed. 2018, 57, 6236–6241; Science 2018, 361, 912–916). **$M^q(CO)_7$ and $M^q(CO)_8$ systems can be both synthesized successfully, suggesting both of them are reasonable species, although $M^q(CO)_7$ ions are more stable.**

But as for the synthesized $M^q(CO)_8$ (Sc^-, Y^-, La^-) anions, which are the 20-e systems. The 20 electrons fulfill ten orbitals including a 'ligand-only' orbital (a_{2u}). So $M^q(CO)_8$ (Sc^-, Y^-, La^-) ions appear as the singlet states (Angew. Chem. Int. Ed. 2018, 57, 6236–6241). Then $M^q(CO)_8$ ($M^q = Ca, Sr$ and Ba) complexes have been synthesized continuously, appearing the similar properties of valence bonding orbitals as the triplet states (Science 2018, 361, 912–916).

So what is the 'ligand-only' orbital? We are curious that whether the ligand-only a_{2u} orbital as a component participate in the eight coordinative bonds to form a 20-electron configuration for its valence electrons. **That is why we begin this research.**

So this manuscript is trying to explain the phenomenon based on a series of experiments. **In our study, we emphasize the 20-e jellium model is an appropriate tool to understand the valence orbital of $M^q(CO)_8$ ($M^q=Ca^{2-}, Sc^-, Ti, V^+, Cr^{2+}$ or Ba^{2-}) complex/ions**, where 20 is the magic number of Jellium model. Furthermore, all the 10 valence orbitals ($a_{1g} + 3t_{1u} + 3t_{2g} + a_{2u} + 2e_g$) can be fulfilled with maximum 20 electrons as the singlet states for the octacarbonyl metal complexes or ions. **That is the significance of this study.**

Therefore, we theoretically design six O_h -symmetric octacarbonyl metal complexes or ions with 20-e on purpose. Actually, if we apply Jellium model, there are no differences of the valence orbitals among the experimental alkaline earth complexes

$M(\text{CO})_8$ ($M = \text{Ca}, \text{Sr}, \text{or Ba}$) with 18-e and the theoretical designed O_h -symmetric $[\text{BaBe}_8]^{2-}$ and $M^q(\text{CO})_8$ ($M^q = \text{Ca}^{2+}, \text{Sc}^+, \text{Ti}, \text{V}^+, \text{Cr}^{2+}$ or Ba^{2+}). All the valence orbitals of such complexes or ions are similar with each other, which are $a_{1g} + t_{1u} + t_{2g} + a_{2u} + e_g$ orbitals. These orbitals can be fulfilled with 20 electrons. The designed analogical comparison of $[\text{BaBe}_8]^{2-}$ and $[\text{Ba}(\text{CO})_8]^{2-}$ is in order to demonstrate Jellium model can be applied to analyse the bonding structure of $[\text{Ba}(\text{CO})_8]^{2-}$. We also emphasize the relevant details in last version manuscript:

On the basis of the experimental researches, all the 10 valence orbitals ($a_{1g} + 3t_{1u} + 3t_{2g} + a_{2u} + 2e_g$) can be fulfilled with maximum 20 electrons for the O_h -symmetric octacarbonyl metal complexes or ions as singlet state. Therefore, it is attractive for us to understand the state of the ligand-only orbital (a_{2u} orbital) essentially.

For another aspect, the octacarbonyl metal complex or ion is with O_h symmetry, where the octa-coordinative field contributed by the positive center metal and eight ligands can be approximately viewed as a homogeneous spherical field or a Jellium model^{12,13}. Therefore, we design a series of O_h -symmetric and singlet-state $M^q(\text{CO})_8$ ($M^q = \text{Ca}^{2+}, \text{Sc}^+, \text{Ti}, \text{V}^+, \text{Cr}^{2+}$ or Ba^{2+}) complex/ions theoretically to learn their bonding structures. Such $M^q(\text{CO})_8$ complex/ions are 20-e closed-shell molecules based on the configuration³, which is exactly satisfying the magic stability of Jellium model^{12,13}.

II) $M^q(\text{CO})_7$ and $M^q(\text{CO})_8$ systems are the stable points on the PESs

We have optimized all the O_h -symmetric $M^q(\text{CO})_8$ ($M^q = \text{Ca}^{2+}, \text{Sc}^+, \text{Ti}, \text{V}^+$ and Cr^{2+}) complex or ions, where no virtual frequencies have been found in the results, indicating the five $M^q(\text{CO})_8$ systems are stable structures on the PESs. The vibrational frequencies of five $M^q(\text{CO})_8$ are listed below (part of Supplementary Table 1).

Compounds/ ions	vibrational frequencies
$[\text{Ca}(\text{CO})_8]^{2-}$	43.30, 43.30, 44.66, 44.66, 44.96, 44.96, 44.96, 51.73, 51.73, 51.73, 58.69, 58.69, 58.69, 197.30, 197.30, 197.30, 207.18, 211.82, 279.90, 279.90, 279.90, 296.30, 296.30, 296.30, 320.23, 320.23, 347.30, 347.30, 347.30, 376.79, 376.79, 376.79, 386.56, 386.56, 386.56, 396.47, 396.47, 1939.24, 1939.24, 1939.24, 1943.70, 1943.70, 1943.70, 1952.20, 2078.08
$[\text{Sc}(\text{CO})_8]^+$	45.03, 45.03, 67.31, 67.31, 67.31, 69.32, 69.32, 72.31, 72.31, 72.31, 73.58, 73.58, 73.58, 226.94, 226.94, 226.94, 244.64, 262.06, 301.11, 301.11, 301.11, 310.73, 310.73, 310.73, 430.24, 430.24, 451.69, 451.69, 451.69, 455.74, 455.74, 455.74, 459.44, 459.44, 468.02, 468.02, 468.02, 2017.66, 2017.66, 2017.66, 2037.30, 2037.30, 2037.30, 2046.58, 2154.28
$\text{Ti}(\text{CO})_8$	45.72, 45.72, 78.65, 78.65, 78.65, 78.09, 78.09, 78.09, 84.15, 84.15, 84.17, 84.17, 84.17, 207.31, 207.31, 207.31, 219.90, 234.97, 234.97, 234.97, 269.58, 286.69, 286.69, 286.69, 471.45, 471.45, 471.45, 477.18, 477.18, 500.36, 500.36, 515.10, 515.10, 516.24, 516.24, 516.24, 2123.43, 2123.43, 2123.43, 2144.97, 2144.97, 2144.97, 2149.43, 2231.66

$[\text{V}(\text{CO})_8]^+$	43.90, 43.90, 74.26, 74.26, 74.26,77.23, 77.23, 77.23, 83.12, 83.12, 83.12, 85.26, 85.26, 115.01, 115.01, 115.01, 135.21, 145.83, 145.83, 145.83, 219.70, 226.80, 226.80, 226.80, 396.68, 396.68, 437.21, 437.21, 437.21, 477.66, 477.66, 477.66, 497.94, 497.94, 505.03, 505.03, 505.03, 2252.88, 2252.88, 2252.88, 2262.96, 2263.33, 2263.33, 2263.33, 2307.85
$[\text{Cr}(\text{CO})_8]^{2+}$	21.03, 37.11, 37.11, 54.83, 54.83, 54.83, 76.24, 76.24, 76.24, 79.18, 79.18, 83.61, 83.61, 83.61, 109.53, 109.53, 109.53, 125.06, 125.06, 125.06, 181.95, 201.80, 201.81, 201.81, 260.40, 260.40, 369.71, 369.71, 369.71, 397.35, 397.35, 397.35, 437.55, 437.55, 437.55, 443.87, 443.87, 2356.57, 2356.66, 2356.66

Moreover, we have found the transition state in the transition of $\text{Ti}(\text{CO})_7 + \text{CO} \rightarrow [\text{OC}\cdots\text{Ti}(\text{CO})_7]^\ddagger \rightarrow \text{Ti}(\text{CO})_8$ (the Figure below). The coordination of the transition state $[\text{OC}\cdots\text{Ti}(\text{CO})_7]^\ddagger$ is listed in the Table. And we can hypothesize this transition should be similar for the other transitions. **The results are also demonstrate both $\text{M}^q(\text{CO})_8$ and $\text{M}^q(\text{CO})_7$ are both the stable points on the PESs, although there is a relative high energy barrier for the transition from $\text{M}^q(\text{CO})_7$ combined with CO.**

So $\text{M}^q(\text{CO})_8$ ($\text{M}^q = \text{Ca}^{2-}, \text{Sc}^-, \text{Ti}, \text{V}^+, \text{Cr}^{2+}$ or Ba^{2-}) are the stable and reasonable species, where Jellium model is a powerful analysis tool in understanding the valence bonding structures of $\text{M}(\text{CO})_8$ or $\text{TM}(\text{CO})_8$ complexes and ions.

Figure: the transition of $\text{Ti}(\text{CO})_7 + \text{CO} \rightarrow [\text{OC}\cdots\text{Ti}(\text{CO})_7]^\ddagger \rightarrow \text{Ti}(\text{CO})_8$ (White: Ti; Grey: C; Red: O)

Table: The coordination of $[\text{Sc}(\text{CO})_8]^-$ under M06-2X/def2tzvpp theoretical level

Atom	x	y	z	Atom	x	y	z
C	1.29387	1.28959	1.22405	O	-1.9471	-1.9515	-1.8766
C	1.22437	-1.3022	1.3019	O	1.95729	-1.8589	-1.9533
C	1.26432	1.26082	-1.2607	O	1.88131	-1.9522	1.95159
C	-1.2972	-1.3015	-1.2198	O	1.95725	1.95302	1.85934
C	-1.2201	1.28945	-1.2896	O	-1.9471	1.87716	1.95098
C	-1.2972	1.2201	1.30123	O	-2.163	-2.166	2.16638
C	1.29388	-1.2238	-1.2897	O	-1.8556	1.95269	-1.9529
C	-1.5151	-1.519	1.51919	O	1.91998	1.91682	-1.9165

Ti	0.0345	0.03057	-0.0306			
----	--------	---------	---------	--	--	--

Moreover, there are still a lot of minor mistakes in the current version, some are listed as follows:

1. On Pg 2, line 3(left column) of the second paragraph, “cations [TM(CO)₈]- (TM = Sc-, Y- and La-)” should be “anions[TM(CO)₈]- (TM = Sc-, Y- and La-)”; line 8 (left column) of the second paragraph, “in the series M(CO)₈ complexes” should be “in the M(CO)₈ complexes”. line 4 (right column) of the second paragraph of “Results”, “with Oh symmetries” should be “with Oh symmetry”.

We have modified as you recommend:

- 1) “cations [TM(CO)₈]- (TM=Sc-, Y- and La-)” has been revised to “anions[TM(CO)₈]- (TM=Sc-, Y- and La-)”: There is an interesting inconsistency in understanding the electronic structures of octacarbonyl complexes M(CO)₈ (M = Ca, Sr and Ba) or anions [TM(CO)₈]- (TM=Sc-, Y- and La-) on the basis of different chemical bonding theories.
- 2) “in the series M(CO)₈ complexes” has been revised to “in M(CO)₈ complexes or ions”: there are eight degenerate coordination bonds and two π-backdonation bonds between metal and ligands in M(CO)₈ complexes or ions
- 3) “with Oh symmetries” has been revised to “with Oh symmetry”: Both of the anions are strictly satisfying the closed-shell ‘20e-superatoms’ with Oh symmetry.

2. On Pg 3, line 2 (left column) of the first paragraph, “showed” should be “shown”; line 12 (left column) of the first paragraph, “indicate” should be “indicates”.

- 1) “showed” has been revised to “shown”: The comparison of the molecular orbitals of [Ba(CO)₈]²⁻ and [Ba(Be)₈]²⁻ are shown in Fig. 1
- 2) “indicate” has been revised to “indicates”: The similarity of [Ba(CO)₈]²⁻ and [BaBe₈]²⁻ indicates we can understand the valence orbitals of [Ba(CO)₈]²⁻ as a ‘20e-superatom’ based on Jellium model.

3. On Pg 4, line 9 (left column) of the first paragraph, “are adopting” should be “adopts” (a lot of such tense mistakes in the context); line 14 (right column) of the second paragraph, “which are exactly the same with before” should be corrected.

- 1) “are adopting” should be “adopts”: All of them have the same valence electron configuration of a_{1g}²t_{1u}⁶t_{2g}⁶a_{2u}²e_g⁴. All the similar mistakes are listed in the Table below.
- 2) “which are exactly the same with before”: However, the |ONs| of Ti are still 0.63 |e|, which equal to the value including the contribution of a_{2u} orbital.

4. On Pg 5, line 5 (left column) of the second paragraph, “bonds” should be “bond”; line 6 (left column) of the second paragraph, “showed” should be “shown”; line 7 (left column) of the second paragraph, “orbital” should be “orbitals”.

- 1) “bonds” should be “bond”: we further compare the bond population of Metal-C bond of ‘20-e’ Ti(CO)₈ (O_h symmetry) with the standard ‘18-e’ Ti(CO)₇ (C_{3v} symmetry)
- 2) “showed” should be “shown”: where each coordinative bond contains 1/8 non-bonding orbitals and 7/8 bonding orbitals shown in Fig. 3
- 3) “orbital” should be “orbitals”: where each coordinative bond contains 1/8 non-bonding orbitals and 7/8 bonding orbitals shown in Fig. 3.

We have correct the mistakes based on the comments. Here are the details, where all the modifications are highlighted in the table

In the old manuscript	In the revised manuscript
Page 1	
such as the seven-coordinate carbonyl cations $[\text{TM}(\text{CO})_7]^+$ ($\text{TM}=\text{V}^+, \text{Nb}^+$ and Ta^+) and the eight-coordinated carbonyl anions $[\text{TM}(\text{CO})_8]^-$ ($\text{TM}=\text{Sc}^-, \text{Y}^-$ and La^-). ³⁻⁵ The recent experimentally produced alkaline earth metal complexes $\text{M}(\text{CO})_8$ ($\text{M} = \text{Ca}, \text{Sr}$ and Ba) are satisfying the DCD model and 18-electron rule	such as the seven-coordinated carbonyl cations $[\text{TM}(\text{CO})_7]^+$ ($\text{TM}=\text{V}^+, \text{Nb}^+$ and Ta^+) and the eight-coordinated carbonyl cations $[\text{TM}(\text{CO})_8]^+$ ($\text{TM}=\text{Sc}^+, \text{Y}^+$ and La^+). ³⁻⁵ However, in the recent synthesized alkaline earth metal complexes $\text{M}(\text{CO})_8$ ($\text{M} = \text{Ca}, \text{Sr}$ and Ba) or anions $[\text{TM}(\text{CO})_8]^-$ ($\text{TM}=\text{Sc}^-, \text{Y}^-$ and La^-), only the valence electrons occupying metal-ligand bonding orbitals are satisfying the DCD model and 18-electron rule,
There is an interesting inconsistency in understanding the electronic structures of octacarbonyl complexes $\text{M}(\text{CO})_8$ ($\text{M} = \text{Ca}, \text{Sr}$ and Ba) or cations $[\text{TM}(\text{CO})_8]^+$ ($\text{TM}=\text{Sc}^+, \text{Y}^+$ and La^+) on the basis of different chemical bonding theories. It should be noted that all the structures adopt O_h symmetry.	There is an interesting inconsistency in understanding the electronic structures of octacarbonyl complexes $\text{M}(\text{CO})_8$ ($\text{M} = \text{Ca}, \text{Sr}$ and Ba) or anions $[\text{TM}(\text{CO})_8]^-$ ($\text{TM}=\text{Sc}^-, \text{Y}^-$ and La^-) on the basis of different chemical bonding theories. It should be noted that all the structures adopt O_h symmetry.
there are eight degenerate coordination bonds and two π -backdonation bonds between metal and ligands in the series $\text{M}(\text{CO})_8$ complexes	there are eight degenerate coordination bonds and two π -backdonation bonds between metal and ligands in $\text{M}(\text{CO})_8$ complexes or ions
As for $\text{TM}(\text{CO})_8$, it has been hypothesized that the eight σ -type orbitals are contributed by both the ligands and the transition metal	As for $\text{TM}(\text{CO})_8$ complex or ion, it has been hypothesized that the eight σ -type orbitals are contributed by both the ligands and the transition metal
Additionally, Zhou's research pointed out that $\text{M}(\text{CO})_8$ complexes or ions adopt O_h symmetry with the valence electron configuration of $a_{1g}^2 t_{1u}^6 t_{2g}^6 a_{2u}^2 e_g^2$	Additionally, Zhou's research pointed out that $\text{M}(\text{CO})_8$ complexes or ions have O_h symmetry with the valence electron configuration of $a_{1g}^2 t_{1u}^6 t_{2g}^6 a_{2u}^2 e_g^2$
the other 9 MOs ($a_{1g} + 3t_{1u} + 3t_{2g} + 2e_g$) including 7 σ -donation bonds ($a_{1g} + 3t_{1u} + 3t_{2g}$) and 2 π -backdonation bonds ($2e_g$) can accommodate 18 valence electrons of $\text{M}(\text{CO})_8$ complex perfectly	the other 9 MOs ($a_{1g} + 3t_{1u} + 3t_{2g} + 2e_g$) including 7 σ -donation bonds ($a_{1g} + 3t_{1u} + 3t_{2g}$) and 2 π -backdonation bonds ($2e_g$) accommodate 18 valence electrons of $\text{M}(\text{CO})_8$ complex perfectly
On the basis of the experimental studies, all the 10 valence orbitals ($a_{1g} + 3t_{1u} + 3t_{2g} + a_{2u} + 2e_g$) can be fulfilled with maximum 20 electrons as the singlet states for the O_h -symmetric octacarbonyl metal complexes or ions.	On the basis of the experimental researches, all the 10 valence orbitals ($a_{1g} + 3t_{1u} + 3t_{2g} + a_{2u} + 2e_g$) can be fulfilled with maximum 20 electrons for the O_h-symmetric octacarbonyl metal complexes or ions as singlet state.
For another aspect, the octacarbonyl metal complex or ion is with O_h symmetries, where the octa-coordination field contributed by the positive center metal and eight ligands can be approximately viewed as a homogeneous spherical field or a Jellium model	For another aspect, the octacarbonyl metal complex or ion is with O_h symmetry, where the octa-coordinative field contributed by the positive center metal and eight ligands can be approximately viewed as a homogeneous spherical field or a Jellium model
Both of the anions are strictly satisfying the closed-shell '20e-superatoms' with O_h symmetries.	Both of the anions are strictly satisfying the closed-shell '20e-superatoms' with O_h symmetry.
It should be noticed that the a_{2u} orbital is an 1F orbital rather than 2S orbital, where the energy of 1F orbital is in the	It should be noticed that the a_{2u} orbital is a 1F orbital rather than the 2S orbital, where the energy level of 1F orbital is in

middle of the two groups 1D orbitals	the middle of the two groups of 1D orbitals
Based on the calculation, a_{2u} orbital adopts the f symmetry as showed in Fig. 1	Based on the calculation, a_{2u} orbital with f symmetry is shown in Fig. 1.
Therefore, the energy of $1F_{xyz}$ orbital is lower than $2S$ orbital caused by the splits of F orbitals	Therefore, the energy of $1F_{xyz}$ orbital is lower than $2S$ orbital caused by the splitting of F orbitals
Page 2	
The comparison of the molecular orbitals of $[\text{Ba}(\text{CO})_8]^{2-}$ and $[\text{Ba}(\text{Be})_8]^{2-}$ are showed in Fig. 1	The comparison of the molecular orbitals of $[\text{Ba}(\text{CO})_8]^{2-}$ and $[\text{Ba}(\text{Be})_8]^{2-}$ are shown in Fig. 1.
The e_g MOs are corresponding to the $1D_{x^2-y^2}$ and $1D_{z^2}$ orbitals as two π -type orbitals, corresponding to the two $d \rightarrow \pi^*$ bonds donated from 5d AOs of Ba ($5d_{x^2-y^2}$ and $5d_{z^2}$ AOs) to the 2p AOs of eight CO groups.	The e_g MOs are corresponding to the $1D_{x^2-y^2}$ and $1D_{z^2}$ orbitals as two π -type orbitals, which are the two $d \rightarrow \pi^*$ bonds donated from 5d AOs of Ba ($5d_{x^2-y^2}$ and $5d_{z^2}$ AOs) to the 2p AOs of eight CO groups
The similarity of $[\text{Ba}(\text{CO})_8]^{2-}$ and $[\text{BaBe}_8]^{2-}$ indicate we can understand the valence orbitals of $[\text{Ba}(\text{CO})_8]^{2-}$ as a '20e-superatom' based on Jellium model.	The similarity of $[\text{Ba}(\text{CO})_8]^{2-}$ and $[\text{BaBe}_8]^{2-}$ indicates we can understand the valence orbitals of $[\text{Ba}(\text{CO})_8]^{2-}$ as a '20e-superatom' based on Jellium model
This is also consistent with the sequence of energies of orbital levels, where the energy of $1F_{xyz}$ orbital as a non-bonding orbital is between 3-degenerate 1D orbitals ($1D_{xy, xz, yz}$) and 2-degenerate 1D orbitals ($1D_{x^2-y^2}$ and $1D_{z^2}$)	This is also consistent with the sequence of molecular orbital level energy, where the energy of $1F_{xyz}$ orbital as the non-bonding orbital is higher than that of ($1D_{xy, xz, yz}$) and lower than the energy of $1D_{x^2-y^2}$ and $1D_{z^2}$ orbitals.
ON s	 ONs
Page 3	
The typical molecules selected as $[\text{Ca}(\text{CO})_8]^{2-}$, $[\text{Sc}(\text{CO})_8]^-$, $\text{Ti}(\text{CO})_8$, $[\text{V}(\text{CO})_8]^+$ and $[\text{Cr}(\text{CO})_8]^{2+}$ all with O_h symmetries are optimized under M06-2X/def2tzvpp theoretical level to obtain their singlet electronic ground states. All of them are adopting the same valence electron configurations of $a_{1g}^2 t_{1u}^6 t_{2g}^6 a_{2u}^2 e_g^4$.	The typical molecules selected as $[\text{Ca}(\text{CO})_8]^{2-}$, $[\text{Sc}(\text{CO})_8]^-$, $\text{Ti}(\text{CO})_8$, $[\text{V}(\text{CO})_8]^+$ and $[\text{Cr}(\text{CO})_8]^{2+}$ with O_h symmetry are optimized under M06-2X/def2tzvpp theoretical level to obtain their singlet electronic ground states. All of them have the same valence electron configuration of $a_{1g}^2 t_{1u}^6 t_{2g}^6 a_{2u}^2 e_g^4$.
The AdNDP analyses are showed in Fig. 2a	The AdNDP analyses are shown in Fig. 2a.
AdNDP analysis reveals seven 9c-2e bonds (including one 1S, three 1P, 3-degenerate 1D and one 1F super orbitals) are corresponding to the eight σ bonds.	AdNDP analysis reveals seven 9c-2e bonds (including one 1S, 3-degenerate 1P, 3-degenerate 1D and one 1F super orbitals) are corresponding to the eight σ bonds
However, the ONs of Ti are 0.63 e , which are exactly the same with before	However, the ONs of Ti are still 0.63 e , which equals to the value including the contribution of a_{2u} orbital
suggest the both the a_{2u} orbitals contribute around 1/8 components in the coordinative orbital	suggests the a_{2u} orbital contribute around 1/8 components in the coordinative orbital of the five $M^q(\text{CO})_8$ systems
Page 5	
we further compare the bond populations of Metal-C bonds of '20-e' $\text{Ti}(\text{CO})_8$ (O_h symmetry) with the standard '18-e' $\text{Ti}(\text{CO})_7$ (C_{3v} symmetry)	we further compare the bond population of Metal-C bond of '20-e' $\text{Ti}(\text{CO})_8$ (O_h symmetry) with the standard '18-e' $\text{Ti}(\text{CO})_7$ (C_{3v} symmetry)
The similar comparison for the other three $M^q(\text{CO})_8$ ions ($M^q = \text{Sc}^-$, V^+ or Cr^{2+}) are listed in the supporting information (SI-table S3)	The similar comparisons for the other three $M^q(\text{CO})_8$ ions ($M^q = \text{Sc}^-$, V^+ or Cr^{2+}) are listed in the supporting information (SI-table S3)
where each coordinative bonds contains 1/8 non-bonding orbital and 7/8 bonding orbital as showed in Fig. 3.	where each coordinative bond contains 1/8 non-bonding orbitals and 7/8 bonding orbitals shown in Fig. 3

In this model, $1D^{10}$ split into two groups including the three low-energy σ-type coordinative orbitals ($1D^6$) and two high-energy π-type backdonation orbitals ($1D^4$). $1F$ super orbital is corresponding to the non-bonding orbital with f symmetry	In this model, $1D^{10}$ orbitals split into two groups including the three σ-type coordinative orbitals ($1D^6$) with lower energy and two π-type backdonation orbitals ($1D^4$) with higher energy. The non-bonding orbital with a_{2u} symmetry is corresponding to $1F$ super orbital ($1F_{xyz}$) rather than the $2S$ orbital, which is caused by the splitting of 1F orbitals
Therefore, 20-e model (Jellium model) is the most appropriate explanation in understanding the valence bonding structures of $M(CO)_8$ or $TM(CO)_8$ complexes and ions	Therefore, 20-e model with the configuration of $1S^2 1P^6 1D^{10} 1F^2$ is the most appropriate explanation for understanding the valence bonding structures of $M(CO)_8$ or $TM(CO)_8$ complexes and ions

REVIEWERS' COMMENTS:

Reviewer #3 (Remarks to the Author):

The authors have addressed all the points, I recommend publication of this manuscript in Communications Chemistry.